# Human brain responses are modulated when exposed to optimized natural images or synthetically generated images

Zijin Gu [1], Keith Jamison [2], Mert R. Sabuncu[1,2] & Amy Kuceyeski [2 ✉]

Understanding how human brains interpret and process information is important. Here, we investigated the selectivity and inter-individual differences in human brain responses to images via functional MRI. In our first experiment, we found that images predicted to achieve maximal activations using a group level encoding model evoke higher responses than images predicted to achieve average activations, and the activation gain is positively associated with the encoding model accuracy. Furthermore, anterior temporal lobe face area (aTLfaces) and fusiform body area 1 had higher activation in response to maximal synthetic images compared to maximal natural images. In our second experiment, we found that synthetic images derived using a personalized encoding model elicited higher responses compared to synthetic images from group-level or other subjects' encoding models. The finding of aTLfaces favoring synthetic images than natural images was also replicated. Our results indicate the possibility of using data-driven and generative approaches to modulate macro-scale brain region responses and probe inter-individual differences in and functional specialization of the human visual system.

[1] School of Electrical and Computer Engineering, Cornell University and Cornell Tech, New York, NY, USA. [2] Department of Radiology, Weill Cornell Medicine, New York, NY, USA. ✉email: amk2012@med.cornell.edu

The brain's visual system has long been a topic of neuroscientific study, with some of the earliest classic psychological models for object recognition being performed over 100 years ago. The identification of preferences in the response patterns of single neurons[1,2] and macro-scale regions[3–5] in the visual cortex has enabled understanding of how brains process and interpret incoming visual information. Artificial neural networks (ANNs), especially deep neural networks, with their architecture motivated by biological neural networks and their striking performance on image classification and object recognition tasks, have naturally lead to their use in modeling the human visual system. Recent work has specifically focused on comparing ANNs trained to predict brain responses from visual stimuli, called encoding models, to the brain's visual system[6–11]. For instance, Kubilius et al. developed a shallow, recurrent ANN representing anatomical brain structures that was shown to accurately reproduce the flow of activity in the primate ventral visual stream[7]. Zhuang et al. found unsupervised ANNs can produce brain-like representations and can achieve accuracy in predicting cortical activations that equals or exceeds best supervised methods[8]; similarly, Mehrer et al. showed ANNs trained on a dataset of brain responses from 1.5 million images across 565 basic categories better predicted representations in higher-level human visual cortex and perceptual judgments than on typical image classification dataset, e.g., ImageNet[9]. While some recent work has highlighted mismatches between ANNs and biological neural networks[12], ANNs remain among the best models for representing and probing visual systems.

Due in part to recent artificial intelligence (AI) breakthroughs in generative models, e.g., generative adversarial networks (GANs)[13], variational autoencoders[14] and diffusion models[15], neural decoding and optimal stimulus design have gained popularity as novel ways to understand and control neural responses to visual stimuli. Coupling pretrained generators with linear or ANN-based encoding models has allowed accurate decoding of viewed images from brain responses that have both high-level semantic and low-level alignment with ground truth[16,17]. ANNs that perform image classification can be coupled with generative networks to synthesize preferred inputs for artificial neurons via activation maximization[18]. Neuroscience researchers have adopted similar approaches for designing optimal stimuli for maximizing firing rate in single neurons or populations of neurons in macaque monkeys[19,20]. Bashivan et al. showed that the firing rate of V4 neural sites can be controlled by a deep ANN as a group, and, to some extent, independently[19]. Ponce et al. revealed the response properties of visual neurons in V1 by exploring the vast generative image space[20]. In terms of human studies, previous work with GAN-based image synthesis showed promising results in testing category selectivity of brain regions and discovering inter- individual and regional difference[21,22]. However, to the best of our knowledge, there is no work thus far that has recorded macro-scale human brain activation in response to synthetic visual stimuli designed to achieve specific, targeted brain activation patterns.

In this work, we aim to enrich our understanding of the human visual system by attempting to modulate activation responses in specific regions of the human brain using selected natural and specifically designed synthetic visual stimuli. We used the large-scale Natural Scences Dataset (NSD)[23], consisting of ~30K coupled images and brain responses from each of eight subjects, to train individual-level ANN-based encoding models with high accuracy. By feeding the NSD images into these encoding models and sorting their predicted average activations, we obtained sets of natural images that were predicted to achieve maximal (or average) levels of activity for that region across the population of NSD subjects. In addition, we used the previously developed NeuroGen framework to design synthetic images predicted to achieve the same goals[22]. Once the natural and synthetic image sets were obtained, we prospectively enrolled six novel individuals and measured their brain responses to these images via functional MRI (fMRI). Once we had image-response data from the six prospectively enrolled subjects, we applied our recently developed linear ensemble method to create personalized, individual-level encoding models for each of these new subjects[24]. With the subjects' personalized encoding models, we then obtained sets of individual-specific natural and synthetic images via the same image selection/generation procedure described in the first experiment, and collected their regional responses to these personalized images during a second fMRI scan. In[24], we validated our approach for building personalized encoding models using small amounts of data, but did not test our framework for targeted activation of given brain regions in specific individuals, which is what we present here. Specifically, we demonstrate that the proposed method can be used to select and generate optimal visual stimuli designed to modulate macro-scale human brain activity in a targeted manner, and, further, that this modulation can be done at the level of a specific individual.

## Results
Figure 1 shows our workflow consisting of the natural image selection and synthetic image generation process for Session 1 and Session 2's fMRI experiments, where Session 1 is on group level and Session 2 is on individual level. Three visual regions, each from a different perception group, i.e., fusiform face area 1 (FFA1), extrastriate body area (EBA) and visual word form area 1 (VWFA1), were identified as primary targets.

**Observed and targeted brain activation patterns are well aligned.** Our Session 1 experimental stimuli were selected based on the group level encoding model trained on NSD as follows. We created eight individual-level encoding models for the above three regions using a deepnet feature-weighted receptive field (deepnet-fwRF) architecture[25]. The deepnet-fwRF model has an ImageNet-pretrained AlexNet[26] to extract features, a Gaussian pooling field to reduce the number of the features and a Ridge regression to map the features to brain regional response, see details in the Methods section. A simple average of the eight individual-level encoding models were taken to obtain a "Group" average encoding model. For natural image selection, the candidate natural images set were the $9000 \times 8 = 72,000$ images shown to any one of the eight NSD subjects while the 1000 images that were shared across subjects were not included. From this set, we selected the top 40 natural images ("Nat") that maximized a region's predicted activation ("Max") for the "Group" encoding model (called "GroupMaxNat") and the 40 natural images ("Nat") that minimized the absolute value of a region's predicted activation, i.e., achieved as closely as possible the average activation response, (called "Avg") for the "Group" encoding model (called "GroupAvgNat"). For synthetic image generation, we inserted the "Group" encoding model into a optimal image synthesis framework called NeuroGen[22] (see details in the Methods section) and generated 40 synthetic images ("Syn") that maximized a region's predicted activation (called "GroupMaxSyn") and 40 synthetic images that minimized a region's absolute predicted activation (called "GroupAvgSyn"). In total, Session 1's image data consisted of 12 stimulus sets (4 sets × 3 regions). And we collected the brain responses from six novel subjects (called NeuroGen subjects) that underwent fMRI while viewing these images.

We began by analyzing Session 1 data on a group level. We fit a linear mixed effects (LME) model with different image conditions as fixed effect and subjects as random effect, and compared the

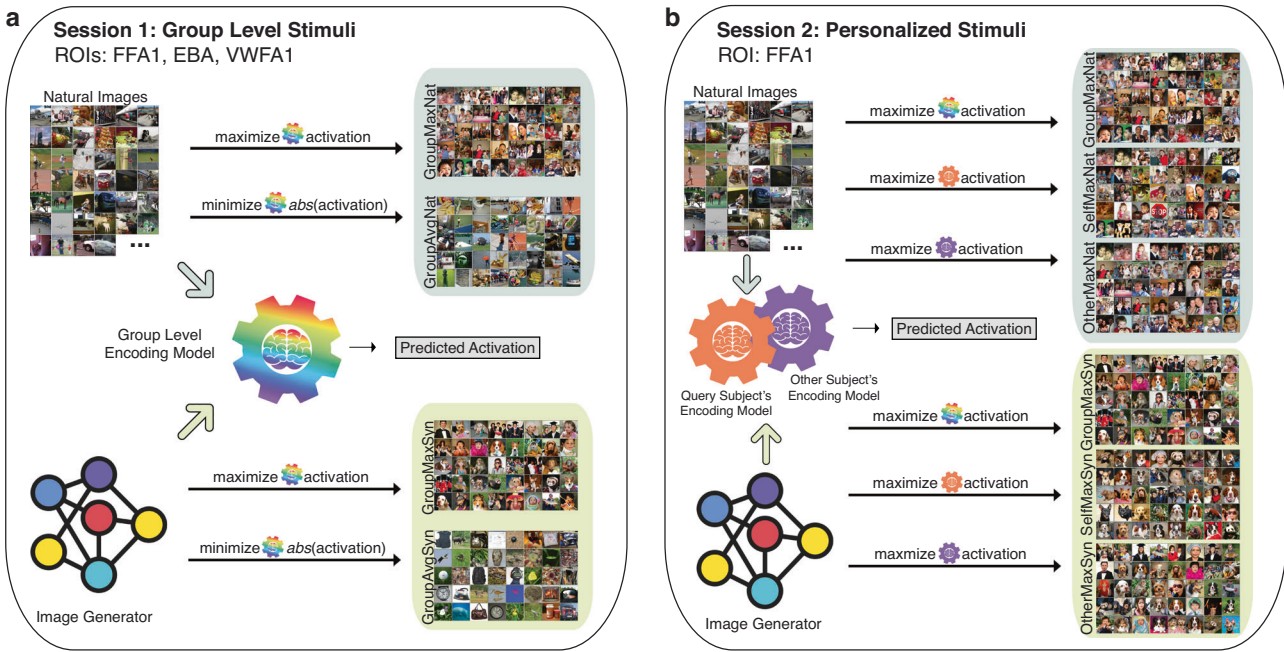

**Fig. 1 Experimental design workflow. a** Session 1: The first experiment involved showing four sets of natural and synthetic images to each of 6 subjects while they underwent fMRI. These sets of images were selected based on their predicted average activation across the 8 NSD subject-level encoding models ("Group" model). Candidate natural images were the set of 9000 × 8 = 72,000 images shown to any subject in the NSD experiments, excepting the shared 1000 images ("Nat"). Synthetic images were created using NeuroGen, which uses BigGAN-deep as its generator ("Syn"). The first set of images, called "GroupMaxNat'', are the natural images with the highest predicted activation in the group NSD encoding model. The second set of images, called "GroupAvgNat'', are the natural images with predicted activations in the NSD group model that are closest to average. The final two sets of images, called "GroupMaxSyn" and "GroupAvgSyn'', are synthetic images designed by NeuroGen to achieve maximal and average activation in the group NSD encoding model, respectively. The regions of interest, or targets, for the session 1 experiments are FFA1, EBA and VWFA1. **b** Session 2: Session 1 data was used to create a personalized encoding model for each of the six subjects, and these personalized encoding models ("Self") were used to select natural and generate synthetic images that were predicted to achieve maximal activation for that person's FFA1 encoding model, named as "SelfMaxNat" and "SelfMaxSyn". During Session 2, we also showed each subject Session 1's group maximal images ("GroupMaxNat" and "GroupMaxSyn") and the other subjects' personalized images ("OtherMaxNat" and "OtherMaxSyn") for FFA1 to test the specificity of the personalization. FFA fusiform face area, EBA extrastriate body area, VWFA visual word form area.

brain activations between image conditions ("GroupMaxNat", "GroupAvgNat", "GroupMaxSyn", "GroupAvgSyn") for all six subjects in the three primary target regions, see Fig. 2a. If the LME coefficient for the condition variable was significant, then those groups were said to have significantly different response levels. We found that "Max" images had significantly higher activity compared to "Avg" images from the same source (natural or synthetic) and no significant differences were found between the "GroupMaxSyn" and "GroupMaxNat" activation responses for FFA1 and EBA, but the natural images have significantly higher responses in VWFA1.

Though inter-regional differences exist, regions belonging to the same perception group (see Supplementary Fig. 1 for the anatomical location of the regions) are usually activated by similar features. Thus, we analyzed all visual regions in the same perception category for stimuli designed for the primary target region from that category. Fig. 2a also shows other face regions' (OFA, FFA1, FFA2, mTLfaces and aTLfaces) activations in response to images designed for FFA1, Fig. 2b shows body regions' (EBA, FBA1, FBA2 and mTLbodies) activations in response to images designed for EBA, and Fig. 2c shows word regions' (OWFA, VWFA1, VWFA2, mfswords and mTLwords) activations in response to images designed for VWFA1. Generally, we observed significantly larger activation in secondary target regions in response to maximal images compared to average images, except in aTLfaces and FBA1 for the natural images and FBA2, mTLbodies, mTLwords for both natural and synthetic images. Finally, maximal synthetic images achieved

significantly higher activations than maximal natural images in secondary target regions aTLfaces and FBA1, while maximal natural images achieved significantly higher activations than maximal synthetic images in the primary word target region VWFA1 and secondary target regions mTLfaces and VWFA2. A detailed version of Fig. 2 showing individual data points is provided in Supplementary Fig. 2. Regional brain responses for off-target images, i.e., body and word region responses to face region-optimized images, are shown in Supplementary Figs. 3–5.

**More accurate encoding models result in better alignment of observed and targeted brain activation patterns.** We hypothesize that the success of our natural or synthetic "Max" images in driving brain activity higher than activity in response to natural or synthetic "Avg" images hinges on the accuracy of the "Group" encoding model for that subject. First, we found that while the "Group" encoding model had a trend toward better accuracy for the natural image responses compared to the synthetic image responses ($t$ statistic = 1.589, $p = 0.114$), overall the prediction accuracies were similar for the two image types (Pearson's $r = 0.467$, $p = 1.652e - 5$), see Supplementary Fig. 6. To test our hypothesis about the relationship between accuracy and success in modulation, we correlated subjects' encoding model accuracy value and the Cohen's $d$ representing activation differences in brain responses to "Max" and "Avg" conditions for natural and synthetic images, see Fig. 3a, b, and c for face, body and word perception groups respectively. Encoding model accuracy was

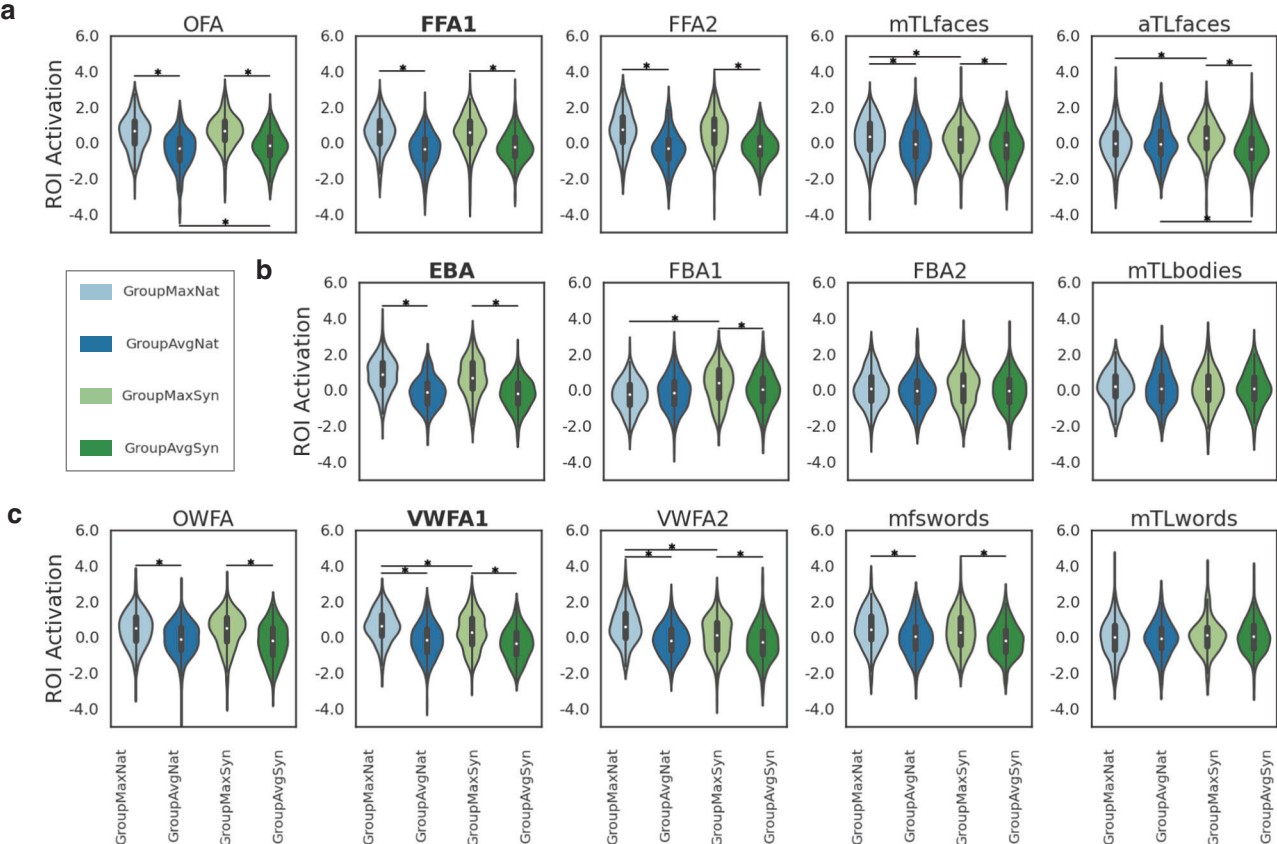

**Fig. 2 Comparisons of brain activations between image conditions in different regions.** Brain responses of the "Max" condition images are significantly higher than the responses of the "Avg" condition images for their targeted region (and for almost all regions in the same response category), for both natural ("Nat") and synthetic ("Syn") image sets. Violinplots show the distribution of the normalized fMRI activity in response to the different image conditions (z-scored over all image responses) in (**a**) face regions, (**b**) body regions and (**c**) word regions. Inside each violin, median is shown by the small white dot in the middle of the boxplot, first quartile and third quartile are indicated by the upper and lower boundary of the black box, and the vertical line shows minimum and maximum. The following group comparisons were performed via fitting linear mixed effects (LME) models: "GroupMaxNat" vs "GroupAvgNat", "GroupMaxSyn" vs "GroupAvgSyn", "GroupMaxSyn" vs "GroupMaxNat" and "GroupAvgSyn" vs "GroupAvgNat". Significant differences based on permutation testing (FDR corrected $p < 0.05$) are marked with a starred horizontal line. Face regions: OFA occipital face area, mTLfaces medial temporal lobe face area, aTLfaces anterior temporal lobe face area. Body regions: FBA fusiform body area, mTLbodies medial temporal lobe body area. Word regions: OWFA occipital word form area, mfswords mid-fusiform sulcus word area, mTLwords medial temporal lobe word area.

determined using all session 1 images that have at least two fMRI measured activations, regardless of condition ("Mat" or "Avg"), source ("Nat" or "Syn") or target region (FFA1, EBA, VWFA1). We found that these correlations were indeed all generally moderate to high positive values, with overall $p$-value < 0.0001 for synthetic stimuli and approximately 0 for natural stimuli, based on permutation test.

In Fig. 4a, we display each of the six subject's 5 most activating natural and synthetic images for FFA1, EBA and VWFA1, sorted in descending order by their measured Session 1 fMRI responses. We observed that while a few of the same top images appear across different subjects, most are not shared across individuals. In Fig. 4b, we performed a pair-wise correlation of each subjects' brain activity responses to quantify inter-subject similarity and found that subjects' responses to the same image vary quite widely, with across-subject correlations ranging from 0 to 0.35. To compare against the noise ceiling, we also include in the diagonal of Fig. 4b the test-retest within-subject reliability in responses to the same image. We see that, in most cases, the diagonal value is larger than the off-diagonal entries.

**Personalized synthetic images allow probing individual differences in brain responses.** We have shown that selecting and

generating images using a group-level encoding model allows targeted modulation of regional brain activity in prospective, novel individuals. Given that there are individual differences in brain responses to images, we hypothesized that selecting and generating personalized natural and synthetic images using an individual-level encoding model might allow more enhanced modulation of regional brain responses. To test this hypothesis, we conducted our second MRI experiment, where we refer to Session 2 data and experimental design workflow is shown in 1b.

Session 2's image set generation followed a similar procedure to Session 1, with the main differences being that the "Group" encoding model was replaced with individual-level, personalized encoding models (called "Self"). We focused only on FFA1, as the face perception regions showed consistent and promising results in group level analysis, see Fig. 2a. The personalized encoding models were constructed via the linear ensemble approach, where the predicted regional activation for the targeted subject is the weighted sum of the predictions from eight NSD FFA1 models plus a bias term[24]. The linear weights and bias were fit using each subject's image-response paired data from Session 1. For each of the six NeuroGen subjects, we created six sets of images and showed them to the subjects. "SelfMaxNat" or "SelfMaxSyn" are sets of natural or synthetic images that maximized FFA1's

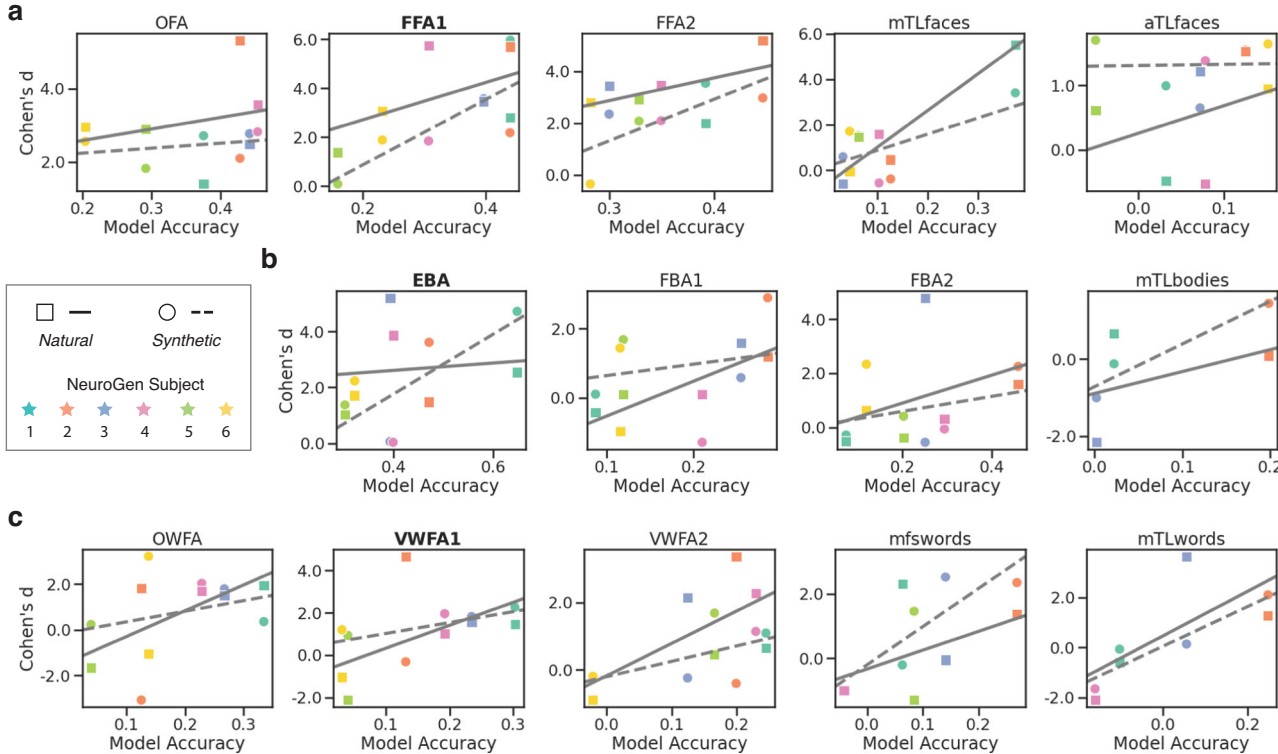

**Fig. 3 Positive correlations between "Max" vs "Avg" activation differences and encoding model accuracy.** The "Max" vs "Avg" activation differences are captured via Cohen's d between activations from "Max" images and "Avg" images. The regional encoding model accuracy is determined using all session 1 images that have at least two fMRI measured activations, regardless of condition ("Mat" or "Avg"), source ("Nat" or "Syn") or target region (FFA1, EBA, VWFA1). The positive associations are observed across the six NeuroGen subjects for all brain regions, where each subject is the same color across all scatter plots. Squares indicate the "Nat" image results where the line of best fit is drawn with a solid line and circles indicate the "Syn" image results where the line of best fit is drawn with a dashed line. Note: some regions were not identifiable in some subjects using the localizer scans. Scatter plots (**a, b**) and (**c**) show the relationship between encoding model accuracy (x-axis) and the Cohen's d calculated by contrasting "GroupMaxNat" vs "GroupAvgNat" or "GroupMaxSyn" vs "GroupAvgSyn" for face, body and word perception regions respectively.

predicted activation for that subject. Besides "Self" condition images, we also showed subjects the personalized FFA1 images of other subjects (called "OtherMaxNat" and "OtherMaxSyn") and the "Max" group level FFA1 images from Session 1 ("GroupMaxNat" and "GroupMaxSyn"). Session 2's fMRI experiments consisted of recording brain responses to 128 images from a total of 6 conditions (32 "SelfMaxSyn", 32 "SelfMaxNat", 20 "OtherMaxSyn", 20 "OtherMaxNat", 12 "GroupMaxSyn" and 12 "GroupMaxNat").

Figure 5b shows, for each of the six subjects, the ten natural and ten synthetic images that had the highest observed responses in FFA1 in descending order. We observed that the top images for different subjects were largely different. Fig. 5a shows the anatomical locations of the five face perception regions of interest where we performed the below comparisons for them to test the effect of the personalization. To test if there was a boost in activation responses from the personalization compared to the group-level images, we compared "GroupMaxSyn" vs "SelfMaxSyn" and "GroupMaxNat" vs "SelfMaxNat". To test the inter-individual specificity of the personalization, we compared "OtherMaxSyn" vs "SelfMaxSyn" and "OtherMaxNat" vs "SelfMaxNat", where "Other" indicates those images are a random subset of the other subjects' personalized image sets. Finally, we tested if the personalized synthetic images had response activations that were higher than the personalized natural images by comparing "SelfMaxNat" vs "SelfMaxSyn". See Supplementary Fig. 7 for the different comparisons between normalized brain activations from different image types.

The comparisons were performed via fitting LME models with different image conditions as fixed effect and subjects as random effect. The results are shown in Table 1 via $\beta$ coefficients from the LME model, where positive means the observed responses to images in the second condition were higher than the observed responses to images in the first condition (and for negative $\beta$, vice versa). Overall, the personalization seemed to work better for the synthetic images compared to the natural images and for the hierarchically later face regions compared to the earlier ones. Specifically, there were higher (though not significant) FFA1 response to the personalized synthetic images compared to the other subjects' personalized synthetic images and higher FFA2 response to the personalized synthetic images compared to the group synthetic images. All three of the later face regions (FFA2, mTLfaces and aTLfaces) had significantly higher (permutation-based corrected $p < 0.05$) activation in response to the personalized synthetic images compared to the other subjects' personalized images, while mTLfaces and aTLfaces also had significantly higher responses to the personalized synthetic images compared to the group synthetic images. The personalization largely had no effect in the natural image responses, with the only significant difference being that the group-level images actually had higher responses than the personalized images for mTLfaces. Finally, while most face regions' responses are not different between the personalized natural and synthetic image sets, similar to what was found using "Group" images, we observe significantly larger responses to synthetic images compared to natural images in the highest-order face region, aTLfaces, and significantly higher

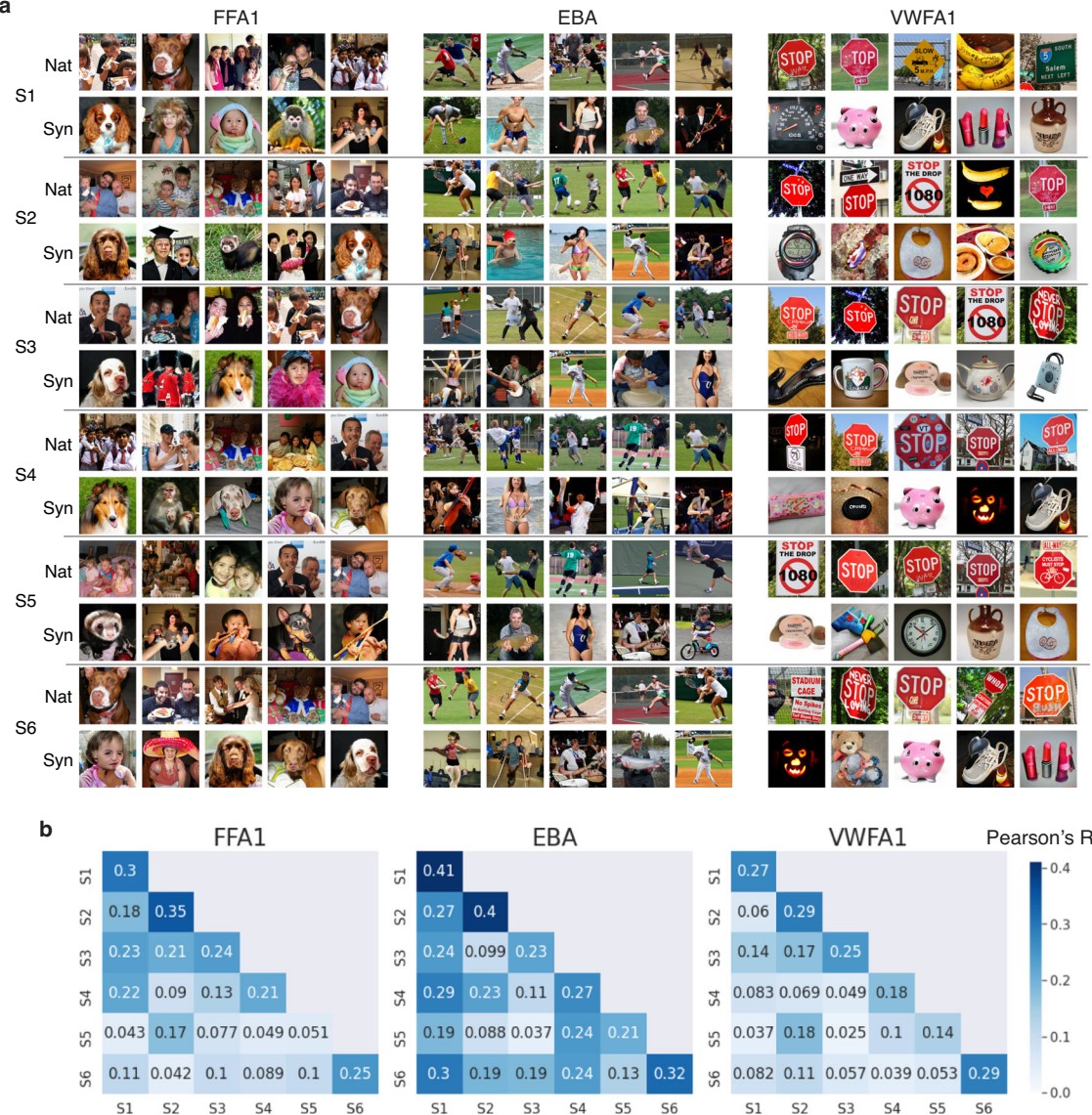

**Fig. 4 Individual differences in brain responses to images. a** For each of the six NeuroGen subjects, the 5 synthetic and 5 natural images that had the highest observed activation in FFA1, EBA and VWFA1. **b** Off-diagonal elements quantify individual differences via across-subject correlation of FFA1, EBA and VWFA1 responses to all images that have two measurements in session 1, while the diagonal elements show within-subject reliability calculated by correlating the two measurements of FFA1, EBA and VWFA1 responses to those images.

responses to natural images compared to synthetic images in the mid-level FFA2 region.

## Discussion

Inspired by previous work in macaques where neuronal firing rates could be driven by optimally designing synthetic images[19,20], here we carried out a set of experiments to show that human brain responses can be modulated in a controlled, personalized way by both selecting optimal natural stimuli and generating optimal synthetic stimuli. All sets of natural or synthetic images designed to maximize activity in targeted brain regions ("Max" conditions) were able to elicit significantly higher observed activity compared to images designed to achieve average activity ("Avg" conditions). Two visual regions, FBA1 and aTL-faces, had significantly higher activation in response to the maximal synthetic images compared to the activation in response to the maximal natural images, while a face area mTLfaces and two word regions VWFA1 and VWFA2 had higher activity in response to natural compared to synthetic images. We also found

that the modulation ability, quantified by Cohen's $d$ between maximal and average brain responses, was associated with the accuracy of the encoding models. That is, more accurate encoding models led to more precise control over brain activity. In addition, inter-individual variability of responses in face regions was considered when creating/selecting the optimal images using an encoding model approach we developed and validated previously[24]. We showed that personalization did indeed drive responses for specific individuals above and beyond the responses to images designed using a group-level encoding model or other individuals' encoding models, but only for synthetic images and only in face regions that were higher in the processing hierarchy. Finally, we observed that, as in Session 1's results, optimal personalized synthetic images had larger responses in the highest-level face processing region aTLfaces, compared to that regions' responses to optimal natural images.

Classically, identifying functional specialization in the brain requires a subject to view a set of images selected by the experimenter based on a priori information or a specific hypothesis of

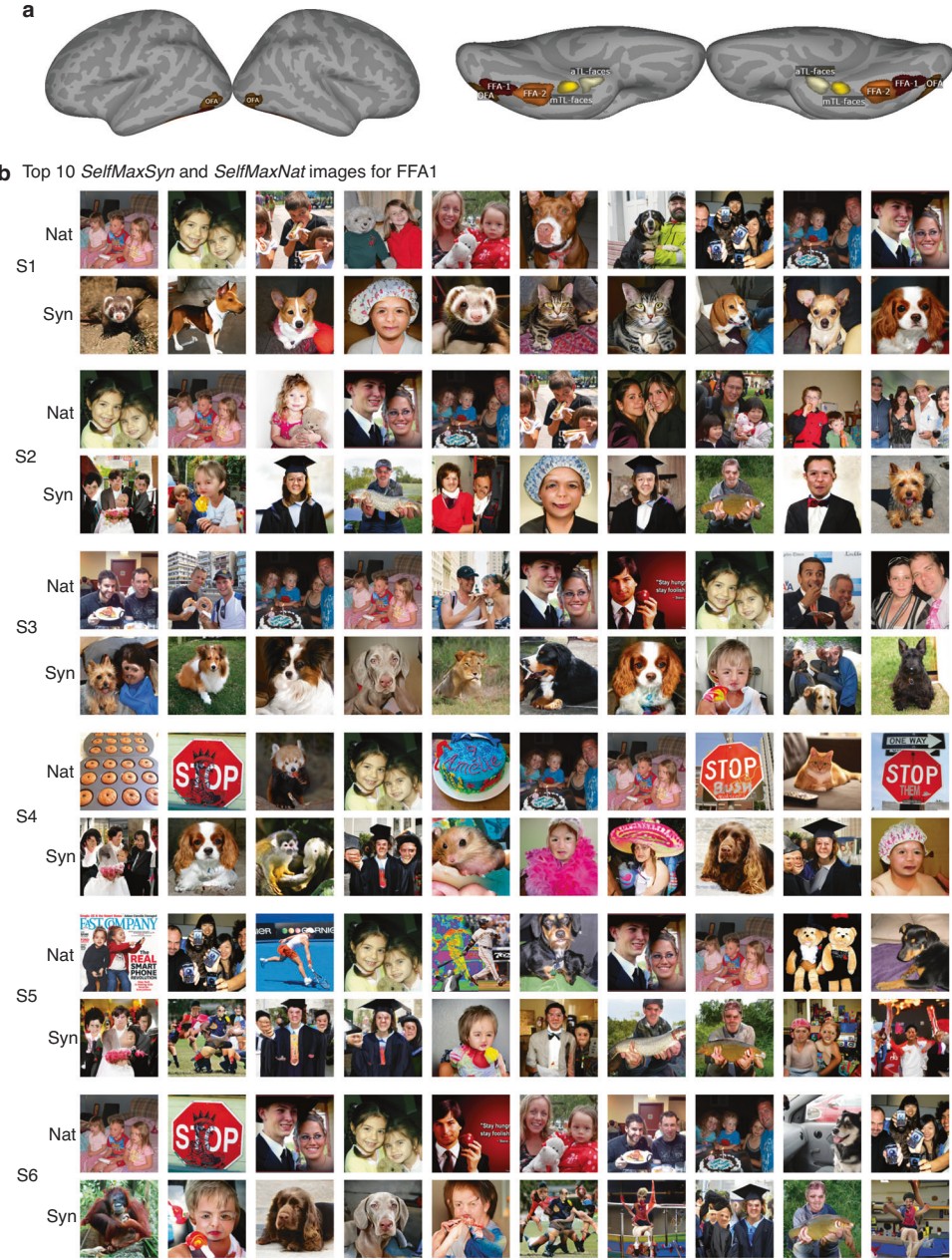

**Fig. 5 The effect of personalization in achieving targeted brain activation patterns in face perception areas. a** The anatomical location of the five face regions in the visual cortex. **b** For each of the six subjects, the ten synthetic and ten natural personalized images that elicited the highest observed FFA1 responses according to the fMRI measurements, in descending order.

the region of interest's preferred image characteristics[27,28]. This type of approach has resulted in identification of various specialized regions in the visual system, including regions that preferentially activate in response to faces[3], bodies[5], places[4], objects[29] and words[30]. However, there are clear limitations to this type of approach in that the content and characteristics of the images are selected by an experimenter with narrow focus, a highly circumscribed hypothesis and limited resources for experimentation. With the monumental progress in generative AI, the publication of large data sets containing image-response information, and improvements in encoding model accuracy using deep learning, the field can and should shift toward using a data-driven approach to selecting and designing optimal stimuli for discovery of functional specialization in the human visual system. This work takes a first step in that direction by robustly

demonstrating the ability to drive responses in various human brain regions using "optimal" selected natural and synthetically generated images, and takes a second step in that direction by showing that personalization of image-response encoding models can allow generation of individual-specific "optimal" images (but perhaps only from synthetic sources).

While ANN-based encoding models may be improved by making them more brain-like[7], the quality and size of the available individual training data is also central to encoding model accuracy. Typically, encoding models require tens of thousands of image-response pairs to obtain good alignment between predictions and observations, such as our NSD-based models that use over 20,000 training samples per subject. Following our recent work[24], here we constructed personalized encoding models using training data that was only ~2% of the NSD data sample size and

**Table 1 The results of fit linear mixed effects (LME) models for assessing personalization effects in face perception regions.**

| contrast | OFA | | FFA1 | | FFA2 | | mTLfaces | | aTLfaces | |
|---|---|---|---|---|---|---|---|---|---|---|
| | β | p-value | β | p-value | β | p-value | β | p-value | β | p-value |
| GroupMaxSyn vs SelfMaxSyn | −0.016 | 0.898 | −0.002 | 0.984 | 0.076 | 0.095 | 0.214 | 0.025[b] | 0.122 | 0.058[a] |
| OtherMaxSyn vs SelfMaxSyn | 0.019 | 0.898 | 0.074 | 0.437 | 0.112 | 0.005[b] | 0.179 | 0.02[b] | 0.134 | 0.025[b] |
| GroupMaxNat vs SelfMaxNat | −0.034 | 0.898 | 0.039 | 0.561 | 0.008 | 0.836 | 0.042 | 0.536 | −0.136 | 0.025[b] |
| OtherMaxNat vs SelfMaxNat | −0.002 | 0.974 | 0.048 | 0.437 | 0.054 | 0.180 | 0.076 | 0.203 | −0.06 | 0.195 |
| SelfMaxNat vs SelfMaxSyn | −0.02 | 0.898 | −0.045 | 0.437 | −0.077 | 0.032[b] | −0.005 | 0.909 | 0.103 | 0.025[b] |

FFA1 is the primary targeted region which is highlighted with bold. Comparisons include "GroupMaxSyn" vs "SelfMaxSyn", "OtherMaxSyn" vs "SelfMaxSyn", "GroupMaxNat" vs "SelfMaxNat", "OtherMaxNat" vs "SelfMaxNat", and "SelfMaxNat" vs "SelfMaxSyn". Columns representing different regions are arranged from posterior (lower-order) to anterior (higher-order). The β values are the model coefficients, where positive values mean the fMRI activation from the second condition is higher than activation from the first condition; corresponding two-tailed p-values were calculated based on permutation testing with FDR corrected per region (p < 0.05).
[a] indicates significant before correction.
[b] indicates significant before and after correction.

obtained relatively good accuracy. However it is likely that more training data for a given subject (particularly synthetic image-response pairs) would increase encoding model accuracy and/or result in models that better reflect inter-individual differences. Furthermore, the performance of the encoding model is also influenced by the images it is built on. In our work, the personalized encoding models were trained and tested on the group level "Max" and "Avg" images for three specific regions. Though we have found no significant difference between the accuracy on natural and synthetic images, there is still bias introduced from the features of the "Max" images, i.e., the encoding model might have better predictions for face, body and text images comparing with images with other features. Since we found that the success of the "optimal" images in hitting their target responses was closely related to encoding model accuracy, we conjecture that increasing the sample size (and more repeats per image to increase SNR) and the diversity of the individual image-response pair training set may lead to better encoding model performance and more precise control over elicited brain responses.

Previous work has mapped the face processing stream in visual cortex, namely with activity flowing from OFA to FFA (sometimes split into FFA1 and FFA2) to medial temporal and then anterior temporal lobe[31]. We observed increasing "success" of the personalized synthetic images when arranging the results from low to high-order (OFA, FFA1, FFA2, mTLfaces to atLfaces), though the targeted region is only FFA1. To formally test this observation, we performed a post-hoc LME analysis with the t statistic between "Self" and "Group"/"Other" as the dependent variable, hierarchical level of region (OFA = 1, FFA1 = 2, FFA2 = 3, mTLfaces = 4, aTLfaces = 5) and the contrast type ("Self" vs "Group", "Self" vs "Other") as fixed effects, and subjects as random effect. We found a positive but not significant coefficient for region hierarchy level (p-value ~ 0.10). It is not surprising that face images designed to activate FFA1 can also activate other face regions, and we did not attempt to design images that also suppressed activation in any of the other off-target regions within the same category (which could be done to increase specificity in the activation responses). The stronger personalization effect in higher order regions could perhaps be explained by more homogeneity in brain responses across the population to low-level characteristics, like facial topology[32], but less homogeneity in brain responses to higher-order characteristics, like facial recognition[33,34]. Furthermore, we observed that the effect of personalization in the natural image responses is much weaker comparing with the effect in the synthetic image responses, which might be due to (1) less inter-subject homogeneity in higher-order face regions' responses to the synthetic images (compared to the natural ones), as they are more novel (and perhaps demand more attention) for higher-order tasks like facial recognition; and the iterative nature of the synthetic image generation that could result in more power to optimize image features that favor the specific subjects than selection from a set of predetermined, fixed natural images.Interestingly, aTLfaces, as the highest order face perception region in our analysis, is the only one that showed consistently significantly higher responses for the synthetic images compared to the natural ones in both group level and individual level experiments, which might be reflecting this phenomena.

Unlike experiments in macaque monkeys where microelectrode arrays can be invasively implanted directly on the brain to record neuronal responses[19,20], human experiments mostly rely on non-invasive recording techniques, with the exception of electrocorticography which is only used in small cohorts of neurological patients[35].While fMRI is one of the best non-invasive methods with which to measure human brain responses as this modality has high spatial resolution (compared to EEG,

not microelectrode arrays) and whole-brain coverage, its main limitations are that it measures non-neuronal BOLD signals, has relatively low temporal resolution and can be subject to imaging artifacts related to acquisition and subject motion. Microelectrode arrays also have the advantage in that they allow recording of neuronal responses in real time so that their firing rates can be monitored in a closed-loop fashion during synthetic image optimization[20]. Real-time fMRI monitoring of brain responses to on-the-fly generated or selected images in an iterative, online optimization would be difficult. With our current configuration, we can only employ post hoc synthesis and offline optimization, which may partly explain the relatively weaker modulation ability in our human study compared with single-cell neuronal modulation studies in animals.

BigGAN-deep, state-of-the art at the time of the experimental design, is an image generator that allows a user-defined balance between fidelity and variety of the synthetic images. Here, we chose more emphasis on fidelity in return for lower variety since we wanted to begin our experiments with images as close as we could to the distribution of natural images used to train the encoding model. Even still, some of the generated images do not look entirely natural. The fact that synthetic maximal image responses in word areas were significantly lower than the responses to the maximal natural images could be attributed to the fact that BigGAN-deep was not trained to produce images with only text[13]. Most of the VWFA1 synthetic images are of items that normally contain text (speedometer, bottles, watches, etc) but the actual text is not readable; the natural images on the other hand contain items with obvious, readable text, e.g., road signs. While the generated faces and bodies are also not completely life-like they are still recognizable as human faces or bodies. Future work, particularly including generative networks that create more accurate rendering of human faces and limbs, e.g., MidJourney v5[36], could further improve the performance of the synthetic images in achieving targeted brain activity. Finally, since we were interested in the effect of the image condition (and not the individual images themselves), we did not include image as a random effect; this could mean that the results may not generalize to out-of-sample images.

The current focus of NeuroGen, as the first step validating its use in human brain modulation, is to target at specific regions which are well-studied and known to be specialized in their response to certain commonly occuring objects in nature, features like face, body and text. It thus stands to reason that natural images would be appropriate for achieving maximal activation in those regions. A perfect implementation of NeuroGen would have a voxel-wise encoding model that could provide accurate prediction of any user-defined pattern of brain responses to images of anything, coupled with a powerful generative model to explore the pixel space and return images with high fidelity and diversity. One can imagine a scenario that involves using this kind of NeuroGen to identify images that maximally activate an arbitrary set of voxels that may not necessarily be activated by the content that is generally contained in natural images. For example, most image sets do not contain obscure or imaginary content, i.e., a zebra riding a bicycle, which could potentially be what maximally activates a certain set of voxels—this kind of content could be created easily using existing generative AI frameworks. In such case, NeuroGen would likely be much more efficient and effective compared to optimal image search over a set of fixed natural images of commonly occurring, realistic content. The general framework used here could also be extended beyond visual cortex to the auditory cortex with audio stimuli, or even the combination of both. Boosting the activation of one region while suppressing the other region is also possible with NeuroGen, which has been proved successful with the artificial brain[22]. Moreover,

comparing with current invasive neuromodulatory techniques, e.g., deep brain stimulation, and non-invasive techniques, e.g., transcranial magnetic, which have limited specificity, NeuroGen may also have broader implications for developing a non-invasive and personalized neuromoduultory method that can be used to functionally target and manipulate brain networks to achieve therapeutic goals.

Taken together, we demonstrate here the possibility of modulating regional human brain responses in a controlled way using group- and individual-level encoding models coupled with either large databases of natural images or generative models that create synthetic images. It appears that achieving targeted control of human brain responses to visual stimuli hinge on three main issues: the accuracy and personalization of encoding models, the content/range of candidate natural image sets and the quality of the image generators. Future directions will focus on improvements in all of these domains, including incorporating semantic content or neuroscientific knowledge into encoding models and using more realistic generators, i.e., stable diffusion[15]. In summary, this approach provides a data-driven method to investigate functional specialization of and possibly a way to modulate regional brain activity in specific humans' brains by either selecting natural or designing synthetic optimal stimuli. We believe this work demonstrates the promise of generating optimal synthetic images, perhaps in the future using better generators, that may succeed in targeted, controlled modulation of brain activations and, in so doing, result in a better understanding of functional specialization within the human visual system.

## Methods

### Data description

*Natural scenes dataset.* The individual encoding models were trained and tested on data from the NSD[23], which contains densely-sampled fMRI data from eight participants (6 female, age 19–32 years). Each subject viewed 9000–10,000 distinct color natural scenes with 2–3 repeats per scene over the course of 30–40 7T MRI sessions (whole-brain gradient-echo EPI, 1.8-mm iso-voxel and 1.6s TR). The images that subjects viewed (3s on and 1s off) were from the Microsoft Common Objects in Context database[37] with a square crop resized to $8.4° \times 8.4°$. Among all images, a set of 1000 were shared across all subjects while the remaining images for each individual were mutually exclusive across subjects. Subjects were asked to fixate centrally and perform a long-term continuous image recognition task (inf-back) to encourage maintenance of attention.

NSD data processing has been previously described[23]. Briefly, the fMRI data were pre-processed to correct for slice time differences and head motion using temporal interpolation and spatial interpolation. Then the single-trial beta weights representing the voxel-wise response to the image presented was estimated using a general linear model (GLM). There are three steps for the GLM: the first is to estimate the voxel-specific hemodynamic response functions; the second is to apply the GLMdenoise technique[38,39] to the single-trial GLM framework[40]; and the third is to use an efficient ridge regression[41] to regularize and improve the accuracy of the beta weights, which represent activation in response to the image. FreeSurfer was used to reconstruct the cortical surface, and both volume- and surface-based versions of the voxel-wise response maps were created. Data from the functional category localizer experiment (fLoc)[42] was used to create contrast maps (voxel-wise $t$-statistics) of responses to specific object categories, and region boundaries were then manually drawn on inflated surface maps by identifying contiguous regions of high contrast in the expected cortical location, and thresholding to include all vertices with contrast $> 0$

within that boundary. Early visual ROIs were defined manually using retinotopic mapping data on the cortical surface. Surface-defined ROIs were projected back to fill in voxels within the gray matter ribbon. Region-wise image responses were then calculated by averaging the voxel-wise beta response maps over all voxels within a given region.

The informed consent for all subjects was obtained by NSD. Our data usage was approved by NSD, and complies with all relevant ethical regulations for work with human participants.

*NeuroGen dataset*. We collected prospective data from six individuals (5 female, age 19–25) over two scans on a 3T GE-MR750 scanner[43], see Fig. 1. The study protocol is approved by an ethical standards committee on human experimentation and written informed consent was obtained from all participants. The first MRI scan included an anatomical T1 (0.9 mm iso-voxel), a functional category localizer (floc) to identify higher-order visual region boundaries (as in the NSD experiments), and, finally, a task fMRI where subjects viewed a fixed set of 480 images. Supplementary Fig. 8 shows the experimental design of the task fMRI. Stimuli were presented for 2 s on and 1 s off, and were organized into blocks for each condition. 8 unique stimuli were presented per block, with one image repeated in each block for use as a one-back behavioral task. To encourage consistent attention, subjects were instructed to maintain fixation on a central dot, and press a button when they observed the repeated stimulus. A single 350-s scan consisted of ten 27-s stimulus blocks with 6 s of rest between blocks. Each session consisted of 7–10 task scans. Stimulus images were square cropped and resized to $8.4° \times 8.4°$ and presented using a Nordic Neuro Lab 32" LCD monitor positioned at the head of the scanner bed. FMRI data consisted of posterior oblique-axial slices oriented to capture early visual areas and the ventral visual stream (gradient-echo EPI, $2.25 \times 2.25 \times 3.00$ mm, 27 interleaved slices, TR = 1.45 s, TE = 32 ms, phase-encoding in the A $\gg$ P direction). EPI susceptibility distortion was estimated using pairs of spin-echo scans with reversed phase-encoding directions[44]. Preprocessing included slice-timing correction with upsampling to 1 s TR, followed by a single-step spatial interpolation combining motion, distortion, and resampling to 2 mm isotropic voxels.

The stimuli used in the task fMRI were 240 natural images selected from the union of all individual-specific images shown to the NSD subjects ($9000 \times 8 = 72,000$) and 240 synthetic images created by NeuroGen[22], a generative framework that can create synthetic images within a given image category. For the first scan's task fMRI, there were total four image conditions for each primary target region (FFA1, EBA, and VWFA1), namely "GroupMaxSyn", "GroupMaxNat", "GroupAvgSyn" and "GroupAvgNat", each containing 40 images (3 regions × 4 conditions x 40 images = 480 images total). The "GroupMaxNat" or "GroupAvgNat" are the natural images that achieve maximal or average predicted activations from the NSD group level encoding model for the region in question, while the "GroupMaxSyn" or "GroupAvgSyn" images are synthetic images optimized using NeuroGen to achieve maximal or average predicted activations from the NSD group level encoding model for the region in question.

During the second MRI scan, the six individuals were shown a set of 128 images (half natural and half synthetic) over six conditions designed for FFA1 (32 "SelfMaxSyn", 32 "SelfMaxNat", 20 "OtherMaxSyn", 20 "OtherMaxNat", 12 "GroupMaxSyn" and 12 "GroupMaxNat"). "GroupMaxSyn" and "GroupMaxNat" images are the same as the images in Session 1 for FFA1. "SelfMaxSyn" and "SelfMaxNat" were images from the "Self" personalized linear ensemble encoding model created for that individual (see details in Method Personalized encoding

model construction section) while "OtherMaxSyn" and "OtherMaxNat" were images from other individuals' "SelfMaxNat" and "SelfMaxSyn" image sets. The fMRI experimental setup and image preprocessing were identical to Session 1.

**Deepnet feature weighted receptive field encoding model**. The encoding model used in this work follows the architecture of the deepnet feature weighted receptive field (deepnet-fwRF) described previously[25]. The deepnet-fwRF model uses AlexNet[26] as a backbone to extract salient features from images. The maximum number of feature maps in each AlexNet layer is set to 512. For layers that have more than 512 feature maps, we calculated the variance of each feature maps for all images in NSD and then selected the top 512 maps that had the highest average variance across the images. Then feature maps that have the same spatial resolution were concatenated, which resulted in three concatenated feature maps with size (256,27,27), (896,13,13) and (1536,1,1). A Gaussian pooling field was applied to the feature maps to further reduce the number of features before the final ridge regression which mapped the features to brain regions' responses, which is the average of the voxel-wise activation maps over that region (a scalar). The hyperparameters, namely the center and radius of the Gaussian pooling field, and the regularization parameter of the Ridge regression, were determined by choosing the combination that gave the best performance on a held-out validation set of 3000 image-response pairs using grid search. Specifically, the candidate feature pooling field centers were spaced 1.4° apart, the candidate radius included 8 log-spaced receptive field sizes between 0.04 and 0.4, and the candidate regularization parameters were 9 log-spaced values between $10^3$–$10^7$.

We trained this model for each region and subject in the NSD dataset. The group level model for each region is constructed by averaging the predictions from the eight NSD subjects' regional encoding models. As our personalized encoding model (see below) is constructed via a linear ensemble approach, where the predicted response for a new subject is the weighted sum of the eight NSD models' predictions, we believe that using a standard average of the 8 subjects' encoding models as the group model provides the most apples-to-apples comparison across the sessions.

**Personalized encoding model construction**. We followed our previously developed approach that allows creating personalized linear ensemble models for novel, prospective individuals using small data[24]. This approach was shown to have a good balance between prediction accuracy and its ability to preserve inter-individual differences in responses. The linear ensemble model linearly combines predictions from a set of base encoding models, which are trained on large data. Here, as in our previous publication, the base models are the deepnet-fwRF[25] encoding models trained on each NSD subjects' data. To fit the linear ensemble model to predict brain responses in the prospective NeuroGen subjects, we trained on a subset of the Session 1 data consisting of 32 randomly chosen image-response pairs from each image condition and each region (total 32 × 4 conditions × 3 regions = 384 images), with the remaining image-response pairs being used to test the personalized encoding model accuracy.

**NeuroGen for optimal image synthesis**. We use here our previously developed NeuroGen framework[22], illustrated in Supplementary Fig. 9, which generates images designed to achieve a user-defined brain activation pattern. Essentially, NeuroGen concatenates an image generator (ImageNet pretrained BigGAN-deep[13]) with an encoding model of human vision. The BigGAN-

deep generator takes as input a one-hot encoded class vector and a noise vector, where the class vector indicates the ImageNet 1000 class and the noise vector is initialized with random values and gets updated during optimization. The output of the generator, which is a image, is used as input to the encoding model which then provides the predicted brain responses to that image. By defining the loss function to capture the match between this predicted brain response and the desired brain response, we can obtain optimized images via two steps: (1) identify the most optimal classes and (2) iteratively optimize the noise vector to produce the image that minimizes the loss. To obtain the optimal classes, we generated 100 images (using 100 different random noise vectors) from each of the 1000 classes in ImageNet. We then feed these $1000 \times 100$ images into the encoding model and compute the average of the predicted regional activation for each of the 1000 image classes (over the 100 images from that class). The optimal classes are those that minimize the loss function. For the "Max" conditions, the loss function was the negative of the predicted activation plus a regularization term on the noise vector; and for the "Avg" conditions, the loss function was the absolute value of the predicted activation plus a regularization term. The "Group" encoding model (the average of the 8 NSD subjects' deepnet-fwRF encoding models) was used to generate Session 1's images while the personalized encoding models (individual-level linear ensembles) were used to generate Session 2's images.

**Linear mixed effects modeling to test for response differences**. We used a LME model to test for statistical differences in the magnitude of the brain responses to different image conditions. Session 1 comparisons were made for contrasts (1) "Group-AvgSyn" vs "GroupMaxSyn", (2) "GroupAvgNat" vs "Group-MaxNat", (3) "GroupMaxNat" vs "GroupMaxSyn" and (4) "GroupAvgNat" vs "GroupAvgSyn". Session 2 comparisons were made for contrasts (1) "OtherMaxSyn" vs "SelfMaxSyn", (2) "GroupMaxSyn" vs "SelfMaxSyn", (3) "OtherMaxNat" vs "Self-MaxNat", (4) "GroupMaxNat" vs "SelfMaxNat", and (5) "Self-MaxNat" vs "SelfMaxSyn". We assume that there is a population effect of a contrast (across all subjects), but each subject is allowed to have its own random deviation. The LME model is defined as

$$\mathbf{y} = \mathbf{X}\boldsymbol{\beta} + \mathbf{Z}\boldsymbol{\alpha} + \epsilon \qquad (1)$$

where $\mathbf{y}$ is the $6n \times 1$ response vector for the observed fMRI responses to the $n$ images (across both conditions) for all 6 subjects, $\mathbf{X}$ is the binary $6n \times 1$ fixed effects vector containing the image condition information for $n$ images across all 6 subjects, $\boldsymbol{\beta}$ is the fixed-effect coefficient, $\mathbf{Z}$ is the binary $6n \times 6$ random effects matrix containing subject information, $\boldsymbol{\alpha}$ is the $6 \times 1$ random-effect coefficient vector, and $\epsilon$ is the error in observations. The $p$-values of the model coefficients were calculated using permutation testing, where we randomly permuted the responses from the two image conditions 1000 times and fit the LME model to get $\hat{\beta}$. The $p$-value for the original $\beta$ is the percent of times the random $\hat{\beta}$ is larger in magnitude than the original $\beta$ (two-sided test).

**Statistics and reproducibility**. In Fig. 2 and Table 1, LME model was used to compare brain regional responses from the generated synthetic and selected natural images, where different image conditions were used as fixed effect and 6 subjects were used as random effects. Significance was determined using permutation testing with FDR method of Benjamini and Hochberg to correct for multiple comparisons. In Fig. 3, the discrepancy in brain responses from different image conditions ("Max" vs "Avg") was measured using Cohen's $d$, and encoding model accuracy was

calculated as the Pearson correlation coefficient between predicted brain responses and measured brain responses from images that were seen at least twice by subject. When comparing the accuracies between natural and synthetic images, $t$ test and Pearson's $r$ was used to determine whether there was a significant difference or significant similarity, respectively. The significance of the relationship between model accuracy and Cohen's $d$ was determined by permutation test where the data from all ROIs were gathered.

**Citation gender diversity statement**. Recent work in several fields of science has identified a bias in citation practices such that papers from women and other minorities are under-cited relative to the number of such papers in the field[45]. Here we sought to proactively consider choosing references that reflect the diversity of the field in thought, form of contribution, gender, and other factors. We obtained predicted gender of the first and last author of each reference by using databases that store the probability of a name being carried by a woman[45]. By this measure (and excluding self-citations to the first and last authors of our current paper), our references contain 8.61% woman(first)/woman(last), 18.71% man/woman, 7.89% woman/man, and 64.79% man/man. This method is limited in that (a) names, pronouns, and social media profiles used to construct the databases may not, in every case, be indicative of gender identity and (b) it cannot account for intersex, non-binary, or transgender people. We look forward to future work that could help us to better understand how to support equitable practices in science.

**Reporting summary**. Further information on research design is available in the Nature Portfolio Reporting Summary linked to this article.

## Data availability

The Natural Scene Dataset is publicly available at http://naturalscenesdataset.org. The NeuroGen Dataset is available at https://figshare.com/articles/dataset/NeuroGen_Dataset/23582403. The source data behind Figures 2 and 3 can be found in Supplementary Data 1 and 2.

## Code availability

NeuroGen code is available at https://github.com/zijin-gu/NeuroGen[46]. Supplementary code is available at https://github.com/zijin-gu/neural-modulation[47].

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

## Acknowledgements
This work was funded by the following grants: R01 NS102646 (A.K.), RF1 MH123232 (A.K.), R01 LM012719 (M.R.S.), R01 AG053949 (M.R.S.), NSF CAREER 1748377 (M.R.S), NSF NeuroNex Grant 1707312 (M.R.S), and Cornell/Weill Cornell Intercampus Pilot Grant (A.K. and M.R.S). The NSD data were collected by Kendrick Kay and Thomas Naselaris under the NSF CRCNS grants IIS-1822683 and IIS-1822929.

## Author contributions
A.K., M.R.S., Z.G. and K.J. conceived and conducted the experiments and interpreted the results, Z.G. additionally analysed the results and carried out the statistical analyses. K.J. additionally processed the imaging data and designed the MRI experiment protocols. Z.G. and A.K. wrote the manuscript. All authors reviewed the manuscript.

## Competing interests
The authors declare no competing interests.
