## [Peer Review File · Communications Biology]

Reviewers' comments:

Reviewer #1 (Remarks to the Author):

In this paper, the authors 1) record fMRI data while participants view natural scenes and synthetic images, 2) fit visual encoding models (both on the individual and group level) to predict the neural responses from the visual stimuli, 3) find natural, and generate synthetic, images that aim to maximize or minimize fMRI responses based on the encoding models, and 4) test how well this works (on the group and individual level). Overall, I think this a very interesting scientific premise, and the authors showed promising initial results for the ability to drive individuals' neural activity with custom synthetic images. I unfortunately found the paper fairly confusing to follow - in my opinion, it turned what should have been an enjoyable paper to read, into a slog to get through. More detailed comments are below.

1. I found the Results section particularly difficult to follow without rereading multiple times (and jumping to intro/methods). At the moment, the beginning of the Results provides an incredible amount of information (including details about the different experiments, many new terms, references to past work, and mentions to things that appear later in the Methods) at once, that the reader then needs to keep clearly in memory in order to understand the remainder of the results. My two general recommendations would be 1) reformat the text so that you provide all the relevant experimental details right before the corresponding results; and 2) before resubmitting, have additional readers, who aren't very familiar with your past work, give you some additional feedback about portions that are challenging to follow. One other location that had a long list describing what you're going to do (which I found challenging to follow) was the beginning of "Personalized synthetic images allow probing individual differences in brain responses" section.

2. Throughout the paper, you conduct many statistical tests without correcting for multiple comparisons, and then (appear to) draw conclusions based on what is significant. Please either correct for multiple comparisons or justify why this is not necessary.

Minor Comments:

-Why do you "Avg" (as in "GroupAvgNat") rather than "Min", if you are finding the image that evokes the minimum response? "Min" would be clearer based on what you're doing (based on my understanding)

-You say that Fig. 2 (in the "Observed and targeted brain activation patterns are well aligned" section) relates to the LME, but it's not clear how in the figure.

-In Fig. 3, is the model accuracy determined on the set of images that includes the "max" images. If so, this analysis seems potentially circular (you will get good prediction of high responses on the max images). I don't think this is the case, as you later on say "We have shown that selecting and generating images using a group-level encoding model allows targeted modulation of regional brain activity in prospective, novel individuals," but this was not clear to me when reading the text corresponding to Fig. 3.

-On page 6, you say that a p-value equals 0. You should either say it's less than the resolution/precision of the permutation test or approximately 0.

-Fig 4b caption: are correlations across all images, or just those showed in panel A?

-In the discussion, you have a sentence "had higher activity in response to natural compared to synthetic regions". "regions" should be "images"

-In the discussion, you give some potential reasons for why the personalized images don't enhance responses (versus the group images) in FFA1 and for natural scenes. In these cases, did the group model do better, or were the individual models more similar between subjects?

-In the methods, you have a sentence "Stimuli were presented for 2s one and 1s off." one -> on

-In the methods, you state that "The group level model for each region is constructed by averaging the predictions from the eight NSD subjects' regional encoding models." If there is a rationale for why you are averaging individual subject models, as opposed to creating a new group model that combines data from all subjects, could you please explain?

Reviewer #2 (Remarks to the Author):

This paper documents an attempt to provide group-based and individualized optimal stimuli based on fMRI-derived encoding models. The paper is clearly-written. The results are modest but not overstated. I'd like to see a few technical clarifications and a bit more of the vision the authors see for this work as it progresses.

What input was given for the class vector in the NeuroGen model? This seems important to understanding the distribution of generated images.

This is incorrect: "Specifically, there were trends for significantly higher FFA1 response to the personalized synthetic images compared to the other subjects' personalized synthetic images and higher..." There cannot be a trend in a single number, and the p-values for each of these comparisons were not significant. Furthermore, it seems the p-values are not corrected for multiple comparisons; why is that?

The individual models were designed to target FFA1, yet this region does not show the expected effects (rather, downstream regions do more so). I'd like to see the authors reflect on this and what it means for the general use of such an approach if the goal is to target activation of a specific area.

The authors say they achieve the "ability to drive responses in various human brain regions in a totally unsupervised way". What is meant by unsupervised here? Supervision is needed to train the individual encoding models. Furthermore, there is probably a good amount of bias that comes in based on what images are used to create the encoding models. The authors should probably speak more to this.

I'd like to see more in the Discussion about what we could learn about the brain if this method were perfected. I'd also like to know if the authors expect such methods to make experiments more efficient. As they note, more individualized training data is likely needed to get a good encoding model. But this may lead to diminishing returns for using NeuroGen to find optimal images if the optimal synthetic image doesn't lead to (much) higher activation than the optimal natural image already shown in the training set (which is the case now).

Reviewer #3 (Remarks to the Author):

In this study, human participants were exposed to selected or synthesized stimuli designed to activate linearized encoding models in one of three visual regions: FFA, EBA, and VWFA1. In general, images chosen or optimized to stimulate specific regions did significantly activate the targeted areas more than images chosen to minimize absolute activation (the "avg" condition) in these regions. Interestingly, selected natural stimuli proved to be equivalent to or more effective than synthetic stimuli.

The primary contribution of this manuscript lies in its method development, demonstrating the application of activation-maximizing in human fMRI, a feat previously shown only in monkey electrophysiology. The neuroscientific contribution is more limited; the only significant neuroscientific finding seems to be an indication of greater inter-subject specificity of the optimal stimuli in higher visual regions. Unless there are other insights that I missed, it appears no additional findings were made about the functional specialization, nature of representations, or mechanisms behind the targeted brain regions. Despite this, I believe that this work has merit for publication as its novel methods could drive scientific discovery in subsequent studies. Below, I list several areas that need improvement before this work is ready for print.

Major comments:

1. I cannot recommend publication without a concrete and committed data-sharing plan for the code, stimuli, and fMRI data. Specifically, the inability to browse the code and the entire stimulus set restricts my capability to thoroughly review this work.

2. Figure 2, a centerpiece of this paper, is difficult to decipher.

2.1. First, no individual observations are depicted. Bar plots, which often obscure more than they reveal, have fallen out of favor in scientific communication.

2.2. Expressing the activation of the maximizing stimuli in standard deviations of the avg stimuli activation results in units that are difficult to interpret, if not devoid of meaning. In theory, the avg condition could consist of multiple copies of the same stimulus. There's nothing in the optimization objective to prevent this. This shows that the standard deviation of the avg stimulus sets is not a meaningful measure of variability for normalizing the max condition. The figure and related analyses should employ a different normalization scheme for the results to be interpretable. Such a normalization scheme could be the percent signal change from baseline or percent signal change from the average activation of the avg condition.

2.3. What about the responses to FFA-targeting stimuli in EBA and VWFA1? The complete activation x targeted region data should be depicted either here or in a supplemental figure.

3. The claim that higher-level regions show greater specificity of the optimized stimuli requires both statistical evidence (i.e., testing a region x personalization interaction, not merely comparing region-related p-values) and a clear link to the existing literature on this topic.

4. As I understand it, the described linear mixed effect model does not consider images as a random factor. In fact, this MLE appears to be a degenerate case equivalent to a paired t-test across subjects, limiting the general applicability of the results.

Minor points:

5. The fact that AlexNet, trained on ImageNet, was used as the basis for the encoding model should be mentioned at the onset of the results section. The absence of model specifics from the results is

curious.

6. A proper permutation test p-value cannot be zero (Phipson & Smyth, 2010 doi: 10.2202/1544-6115.1585).

7. The statement, "human experiments obviously must rely on non-invasive recording techniques," does not take electrocorticography into account.

8. The phrase, "Then the number of features in each AlexNet layer were reduced by selecting those that had the highest variance," is ambiguous. Please precisely describe the feature selection procedure and the method for its cross-validation.

Response to Reviewers

**Title: Modulating human brain responses via optimal natural image selection
and synthetic image generation**

**Manuscript Reference Number:
COMMSBIO-23-1582-T**

Authors:

Zijin Gu

Keith Jamison

Mert Sabuncu

Amy Kuceyeski

Date: July 30, 2023

Message from the Authors

Dear Reviewers,

We thank the reviewers for their constructive comments, which have allowed us to improve the quality of the manuscript. We have addressed the comments and incorporated these valuable suggestions in the current manuscript which is a substantial revision; we believe that the result is a strong manuscript that will be of great interest to the neuroscientific community, vision researchers in particular. The updated contents are colored in blue in the revised manuscript to indicate our changes.

We have made many major changes, including addressing all statistical concerns, incorporating corrections for multiple comparisons, adding discussion related to the underlying models used in the study, and reformatting the text to increase the clarity of the presentation. All page and figure numbers in our response are based on the revised manuscript, unless otherwise stated. The page and reference numbers mentioned in the reviewers' comments are kept intact and are based on the original manuscript. The references contained in this reviewer response document are in author-year format for ease of reading, and are listed at the end of this document. Thank you and we look forward to hearing the journal's decision.

Sincerely,

Zijin Gu, Keith Jamison, Mert Sabuncu, Amy Kuceyeski

Response To Reviewer #1

Overall Comments

In this paper, the authors 1) record fMRI data while participants view natural scenes and synthetic images, 2) fit visual encoding models (both on the individual and group level) to predict the neural responses from the visual stimuli, 3) find natural, and generate synthetic, images that aim to maximize or minimize fMRI responses based on the encoding models, and 4) test how well this works (on the group and individual level). Overall, I think this a very interesting scientific premise, and the authors showed promising initial results for the ability to drive individuals' neural activity with custom synthetic images. I unfortunately found the paper fairly confusing to follow - in my opinion, it turned what should have been an enjoyable paper to read, into a slog to get through. More detailed comments are below.

Response

Thank you for your interest in our paper. We appreciate your careful review and detailed feedback; we have greatly reformatted the manuscript and clearly re-written some sections. We hope you find the revised manuscript easier to read and follow. Please also see our point-by-point responses below.

Reviewer Comment

I found the Results section particularly difficult to follow without rereading multiple times (and jumping to intro/methods). At the moment, the beginning of the Results provides an incredible amount of information (including details about the different experiments, many new terms, references to past work, and mentions to things that appear later in the Methods) at once, that the reader then needs to keep clearly in memory in order to understand the remainder of the results. My two general recommendations would be 1) reformat the text so that you provide all the relevant experimental details right before the corresponding results; and 2) before resubmitting, have additional readers, who aren't very familiar with your past work, give you some additional feedback about portions that are challenging to follow. One other location that had a long list describing what you're going to do (which I found challenging to follow) was the beginning of "Personalized synthetic images allow probing individual differences in brain responses" section.

Response

We have reorganized the Results section, including adding necessary details and removing redundant information, providing the relevant experimental details right before the corresponding results. Particularly, we moved the description of our Session 2 data collection to the beginning "Personalized synthetic images allow probing individual differences in brain responses" section so readers can have a smooth transition from methods development to results. However, as *Communications Biology*

requires Methods to be listed at the end of the manuscript file, we are not able to move the entire Methods section before the Results. I hope you find the updated manuscript easier to read.

Reviewer Comment

Throughout the paper, you conduct many statistical tests without correcting for multiple comparisons, and then (appear to) draw conclusions based on what is significant. Please either correct for multiple comparisons or justify why this is not necessary.

Response

We have corrected the p -values from the statistical tests for multiple comparison with Benjamini/Hochberg FDR corrections. And we have also changed the figures and texts to reflect the correction accordingly. Please see the updated manuscript for details; while there are some changes, largely the results hold with the corrected p -values.

Reviewer Minor Comment

Why do you “Avg” (as in “GroupAvgNat”) rather than “Min”, if you are finding the image that evokes the minimum response? “Min” would be clearer based on what you’re doing (based on my understanding)

Response

The fMRI responses to images can have either positive or negative values reflecting the change in activity elicited by the image presentation. Thus the "Avg" images were designed to minimize the absolute value of the activation of a targeted region (to achieve zero activation), not to achieve minimal (maximal negative) activation. We have modified the text to clarify this point:

From this set, we selected the top 40 natural images ("Nat") that maximized a region's predicted activation ("Max") for the "Group" encoding model (called "GroupMaxNat") and the 40 natural images ("Nat") that minimized the absolute value of a region's predicted activation, i.e. achieved as closely as possible the average activation response, (called "Avg") for the "Group" encoding model (called "GroupAvgNat").

Reviewer Minor Comment

You say that Fig. 2 (in the “Observed and targeted brain activation patterns are well aligned” section) relates to the LME, but it’s not clear how in the figure.

Response

LME models were fit per region for the following comparisons: "GroupMaxNat" vs "GroupAvgNat", "GroupMaxSyn" vs "GroupAvgSyn", "GroupMaxSyn" vs "GroupMaxNat" and "GroupAvgSyn" vs "GroupAvgNat". If the model coefficients for the condition variable were significant, then the conditions were said to have significantly different response levels. We have modified the text to better describe this procedure. As the previous barplots were not informative, we have changed them to violin plots, as shown in Figure 2 below.

Figure 2: Brain activations in response to the "Max" condition images are significantly higher than the activations in response to the "Avg" condition images for their targeted region (and for almost all regions in the same response category), for both natural ("Nat") and synthetic ("Syn") image sets. Violinplots show the distribution of the normalized fMRI activity in response to the different image conditions in **a** face regions, **b** body regions and **c** word regions. Different subjects are represented by different color points. The following group comparisons were performed via fitting linear mixed effects (LME) models: "GroupMaxNat" vs "GroupAvgNat", "GroupMaxSyn" vs "GroupAvgSyn", "GroupMaxSyn" vs "GroupMaxNat" and "GroupAvgSyn" vs "GroupAvgNat". Significant differences based on permutation testing (FDR corrected $p < 0.05$) are marked with a starred horizontal line.

We also added the following sentence to the results section to clarify how we are using the LME model to determine significance.

If the LME coefficient for the condition variable was significant, then those groups were

said to have significantly different response levels.

Reviewer Minor Comment

In Fig. 3, is the model accuracy determined on the set of images that includes the “max” images. If so, this analysis seems potentially circular (you will get good prediction of high responses on the max images). I don’t think this is the case, as you later on say "We have shown that selecting and generating images using a group-level encoding model allows targeted modulation of regional brain activity in prospective, novel individuals," but this was not clear to me when reading the text corresponding to Fig. 3.

Response

The model accuracy shown in Figure 3 was determined using all NeuroGen session 1 images that have at least two fMRI measured activations, which include both "Max" images and "Avg" images designed for all brain regions. We have modified the caption of Figure 3 and also pasted below:

Positive correlations between "Max" vs "Avg" activation differences and encoding model accuracy. The "Max" vs "Avg" activation differences are captured via unpaired t-statistics between activations from "Max" images and "Avg" images. The regional encoding model accuracy is determined using all session 1 images that have at least two fMRI measured activations, regardless of condition ("Mat" or "Avg"), source ("Nat" or "Syn") or target region (FFA1, EBA, VWFA1). The positive associations are observed across the six NeuroGen subjects for all brain regions, where each subject is the same color across all scatter plots. Squares indicate the "Nat" image results where the line of best fit is drawn with a solid line and circles indicate the "Syn" image results where the line of best fit is drawn with a dashed line. Note: some regions were not identifiable in some subjects using the localizer scans. Scatter plots **a**, **b** and **c** show the relationship between encoding model accuracy (x-axis) and the t-statistic calculated by contrasting "GroupMaxNat" vs "GroupAvgNat" or "GroupMaxSyn" vs "GroupAvgSyn" for face, body and word perception regions respectively.

We also added the following text to the section where we discuss the figure:

Encoding model accuracy was determined using all session 1 images that have at least two fMRI measured activations, regardless of condition ("Mat" or "Avg"), source ("Nat" or "Syn") or target region (FFA1, EBA, VWFA1).

Reviewer Minor Comment

On page 6, you say that a p-value equals 0. You should either say it’s less than the resolution/precision of the permutation test or approximately 0.

Response

We have changed the text to "approximately 0".

Reviewer Minor Comment

Fig 4b caption: are correlations across all images, or just those showed in panel A?

Response

The correlations were calculated across all images that have at least two fMRI measured activations in Session 1. To be clear, we modified the caption to

Off-diagonal elements quantify individual differences via across-subject correlation of FFA1, EBA and VWFA1 responses to all images that have two measurements in session 1, while the diagonal elements show within-subject reliability calculated by correlating the two measurements of FFA1, EBA and VWFA1 responses to those images.

Reviewer Minor Comment

In the discussion, you have a sentence “had higher activity in response to natural compared to synthetic regions”. “regions” should be “images”

Response

We apologize for this typo and have corrected it.

Reviewer Minor Comment

In the discussion, you give some potential reasons for why the personalized images don't enhance responses (versus the group images) in FFA1 and for natural scenes. In these cases, did the group model do better, or were the individual models more similar between subjects?

Response

We added another violin plots of the normalized brain activations of Session 2 in Supplementary Figure 9, and also attached below.

We also provided the statistics in Table 1 in the main manuscript shows comparisons between group image responses (based on the group model) and individual image responses to both natural and synthetic images. From the table we see that group model achieves no significant differences in FFA1 activation compared to the individual model. Also, images from other subjects' model achieve similar activation level as individual model, which can be an indication that individual models are similar across subjects in FFA1 region.

Reviewer Minor Comment

Supplementary Figure 9: Responses of FFA1-targeted images in different face regions of Session 2.

In the methods, you have a sentence “Stimuli were presented for 2s one and 1s off.”
one -> on

Response

We apologize for this typo and have corrected it.

Reviewer Minor Comment

In the methods, you state that “The group level model for each region is constructed by averaging the predictions from the eight NSD subjects’ regional encoding models.” If there is a rationale for why you are averaging individual subject models, as opposed to creating a new group model that combines data from all subjects, could you please explain?

Response

We have added the following text to the methods section:

As our personalized encoding model (see below) is constructed via a linear ensemble approach, where the predicted response for a new subject is the weighted sum of the eight NSD models’ predictions, we believe that using a standard average of the 8 subjects’ encoding models as the group model provides the most apples-to-apples comparison across the sessions.

From previous experience, we do not believe that a model trained on a union of all 8 subjects’ data would be very different from averaging the 8 subjects’ encoding model outputs.

Response To Reviewer #2

Overall Comments

This paper documents an attempt to provide group-based and individualized optimal stimuli based on fMRI-derived encoding models. The paper is clearly-written. The results are modest but not overstated. I'd like to see a few technical clarifications and a bit more of the vision the authors see for this work as it progresses.

Response

We would like to thank you for your positive feedback. Your detailed comments have considerably helped to improve the quality of the revised manuscript. We hope you find our below responses satisfactory.

Reviewer Comment

What input was given for the class vector in the NeuroGen model? This seems important to understanding the distribution of generated images.

Response

We have added the following text to clarify how the class vector was selected.

By defining the loss function to capture the match between this predicted brain response and the desired brain response, we can obtain optimized images via two steps: 1) identify the most optimal classes and 2) iteratively optimize the noise vector to produce the image that minimizes the loss. To obtain the optimal classes, we generated 100 images (using 100 different random noise vectors) from each of the 1000 classes in ImageNet. We then feed these 1000 x 100 images into the encoding model and compute the average of the predicted regional activation for each of the 1000 image classes (over the 100 images from that class). The optimal classes are those that minimize the loss function.

Reviewer Comment

This is incorrect: "Specifically, there were trends for significantly higher FFA1 response to the personalized synthetic images compared to the other subjects' personalized synthetic images and higher..." There cannot be a trend in a single number, and the p-values for each of these comparisons were not significant. Furthermore, it seems the p-values are not corrected for multiple comparisons; why is that?

Response

We have changed the sentence with more accurate wording:

Specifically, there were higher (though not significant) FFA1 response to the personalized synthetic images compared to the other subjects' personalized synthetic images and higher FFA2 response to the personalized synthetic images compared to the group synthetic images.

We have modified the table with FDR corrected p -values. Please find the table attached below.

contrast	OFA		FFA1		FFA2		mTLfaces		aTLfaces	
	β	p -value	β	p -value	β	p -value	β	p -value	β	p -value
GroupMaxSyn vs SelfMaxSyn	-0.016	0.898	-0.002	0.984	0.076	0.095	0.214	0.025**	0.122	0.058*
OtherMaxSyn vs SelfMaxSyn	0.019	0.898	0.074	0.437	0.112	0.005**	0.179	0.02**	0.134	0.025**
GroupMaxNat vs SelfMaxNat	-0.034	0.898	0.039	0.561	0.008	0.836	0.042	0.536	-0.136	0.025**
OtherMaxNat vs SelfMaxNat	-0.002	0.974	0.048	0.437	0.054	0.180	0.076	0.203	-0.06	0.195
SelfMaxNat vs SelfMaxSyn	-0.02	0.898	-0.045	0.437	-0.077	0.032**	-0.005	0.909	0.103	0.025**

Table 1: The results of fit linear mixed effects (LME) models for assessing personalization effects in face perception regions by comparing: "GroupMaxSyn" vs "SelfMaxSyn", "OtherMaxSyn" vs "SelfMaxSyn", "GroupMaxNat" vs "SelfMaxNat", "OtherMaxNat" vs "SelfMaxNat", and "SelfMaxNat" vs "SelfMaxSyn". Columns representing different regions are arranged from posterior (lower-order) to anterior (higher-order). The β values are the model coefficients, where positive values mean the fMRI activation from the second condition is higher than activation from the first condition; corresponding two-tailed p -values were calculated based on permutation testing with FDR corrected per region ($p < 0.05$). "*" indicates significant before correction and "**" indicates significant before and after correction.

Reviewer Comment

The individual models were designed to target FFA1, yet this region does not show the expected effects (rather, downstream regions do more so). I'd like to see the authors reflect on this and what it means for the general use of such an approach if the goal is to target activation of a specific area.

Response

We have the following explanations in the Discussion section regarding why there may more success of the maximizing images in the downstream regions:

We observed increasing "success" of the personalized synthetic images when arranging the results from low to high-order (OFA, FFA1, FFA2, mTLfaces to atLfaces), though the targeted region is only FFA1. It is not surprising that face images designed to activate FFA1 can also activate other face regions, and we did not attempt to design images that also suppressed activation in any of the other off-target regions within the same category (which could be done to increase specificity in the activation responses). The stronger personalization effect in higher order regions could perhaps be explained by that there is more homogeneity in brain responses across the population to low-level characteristics of face images, like facial topology (Henriksson et al., 2015), but less homogeneity in brain responses to higher-order characteristics, like facial recognition (Barense et al., 2010; Yang et al., 2016).

The current focus of this work is to elicit maximal activity in a single targeted brain region, as it is the simplest, most straightforward way to validate this novel technology. NeuroGen has more broader use cases as it gets improved, and we have added some text discussing the general usage of NeuroGen:

The current focus of NeuroGen, as the first step in human brain modulation, is to target at specific regions, e.g., FFA, EBA and VWFA, which are well-studied visual regions that specialized for recognition of certain predetermined features like face, body and text. However, NeuroGen, with proper modification on the encoding model and the generative model, can be used in a more general purpose for identifying brain regions and investigating stimuli-responses relationship. For example, with a voxel-wise encoding model, we can target at a single voxel or any group of voxels with any predefined pattern and obtain preferred stimuli for them. This can also be extended beyond visual cortex to the auditory cortex with audio stimuli, or even the combination of both. Boosting the activation of one region while suppressing the other region is also possible with NeuroGen, which has been proved successful with artificial brain (Gu et al., 2022). Moreover, comparing with current invasive neuromodulatory techniques, e.g., deep brain stimulation, and non-invasive techniques, e.g., transcranial magnetic, which have limited specificity, NeuroGen can also have broader implications for developing a non-invasive and personalized neuromodulatory method that can be used to functionally target and manipulate brain networks to achieve therapeutic goals.

Reviewer Comment

The authors say they achieve the "ability to drive responses in various human brain regions in a totally unsupervised way". What is meant by unsupervised here? Supervision is needed to train the individual encoding models. Furthermore, there is probably a good amount of bias that comes in based on what images are used to create the encoding models. The authors should probably speak more to this.

Response

We have removed the phrase "in a totally unsupervised way". We wanted to say that our optimization doesn't involve real-time feedback from subject's observed brain responses like previous works do, so the images are generated without the direct supervision from the subject. And we agree that might cause confusion especially in machine learning setting.

We have added in the Discussion about the bias from the images in creating the encoding model:

Furthermore, the performance of the encoding model is also influenced by the images it is built on. In our work, the personalized encoding models were trained and tested on the group level "Max" and "Avg" images for three specific regions. Though we have found no significant difference between the accuracy on natural and synthetic images, there is still bias introduced from the features of the "Max" images, i.e., the encoding model might have better predictions for face, body and text images comparing with images with other features. Since we found that the success of the "optimal" images in hitting their target responses was closely related to encoding model accuracy, we conjecture that increasing the sample size (and more repeats per image to increase SNR) and the diversity of the individual image-response pair training set may lead to better encoding model performance and more precise control over elicited brain responses.

Reviewer Comment

I'd like to see more in the Discussion about what we could learn about the brain if this method were perfected. I'd also like to know if the authors expect such methods to make experiments more efficient. As they note, more individualized training data is likely needed to get a good encoding model. But this may lead to diminishing returns for using NeuroGen to find optimal images if the optimal synthetic image doesn't lead to (much) higher activation than the optimal natural image already shown in the training set (which is the case now).

Response

We have added the following text to the discussion.

The current focus of NeuroGen, as the first step validating its use in human brain modulation, is to target at specific regions which are well-studied and known to be specialized in their response to certain commonly occurring objects in nature, features like face, body and text. It thus stands to reason that natural images would be appropriate for achieving maximal activation in those regions. A perfect implementation of NeuroGen would have a voxel-wise encoding model that could provide accurate prediction of any user-defined pattern of brain responses to images of anything, coupled with a powerful generative model to explore the pixel space and return images with high fidelity and diversity. One can imagine a scenario that involves using this kind of NeuroGen to identify images that maximally activate an arbitrary set of voxels that may not necessarily be activated by the content that is generally contained in natural images. For example, most image sets do not contain obscure or imaginary content, i.e. a zebra riding a bicycle, which could potentially be what maximally activates a certain set of voxels - this

kind of content could be created easily using existing generative AI frameworks. In such case, NeuroGen would likely be much more efficient and effective compared to optimal image search over a set of fixed natural images of commonly occurring, realistic content.

Response To Reviewer #3

Overall Comments

In this study, human participants were exposed to selected or synthesized stimuli designed to activate linearized encoding models in one of three visual regions: FFA, EBA, and VWFA1. In general, images chosen or optimized to stimulate specific regions did significantly activate the targeted areas more than images chosen to minimize absolute activation (the "avg" condition) in these regions. Interestingly, selected natural stimuli proved to be equivalent to or more effective than synthetic stimuli.

The primary contribution of this manuscript lies in its method development, demonstrating the application of activation-maximizing in human fMRI, a feat previously shown only in monkey electrophysiology. The neuroscientific contribution is more limited; the only significant neuroscientific finding seems to be an indication of greater inter-subject specificity of the optimal stimuli in higher visual regions. Unless there are other insights that I missed, it appears no additional findings were made about the functional specialization, nature of representations, or mechanisms behind the targeted brain regions. Despite this, I believe that this work has merit for publication as its novel methods could drive scientific discovery in subsequent studies. Below, I list several areas that need improvement before this work is ready for print.

Response

Thank you for your recognition of our work. Your suggestions have greatly helped improve our work and please find the point-to-point responses below.

Reviewer Comment

I cannot recommend publication without a concrete and committed data-sharing plan for the code, stimuli, and fMRI data. Specifically, the inability to browse the code and the entire stimulus set restricts my capability to thoroughly review this work.

Response

We definitely agree that making the dataset/code publicly available would be better for the research community. We have our code publicly available at <https://github.com/zijin-gu/neural-modulation/tree/main>. We plan to upload the full set of fMRI data to figshare once we obtain approval from the study's current Institutional Review Board (IRB) for Human Participant Research. Meanwhile, we plan to upload the stimuli first to figshare at https://figshare.com/articles/dataset/NeuroGen_Dataset/23582403.

Reviewer Comment

Figure 2, a centerpiece of this paper, is difficult to decipher. 2.1. First, no individual observations are depicted. Bar plots, which often obscure more than they reveal, have fallen out of favor in scientific communication.

2.2. Expressing the activation of the maximizing stimuli in standard deviations of the avg stimuli activation results in units that are difficult to interpret, if not devoid of meaning. In theory, the avg condition could consist of multiple copies of the same stimulus. There's nothing in the optimization objective to prevent this. This shows that the standard deviation of the avg stimulus sets is not a meaningful measure of variability for normalizing the max condition. The figure and related analyses should employ a different normalization scheme for the results to be interpretable. Such a normalization scheme could be the percent signal change from baseline or percent signal change from the average activation of the avg condition.

2.3. What about the responses to FFA-targeting stimuli in EBA and VWFA1? The complete activation x targeted region data should be depicted either here or in a supplemental figure.

Response

We have changed the bar plots to violin plots in the main text, where we now normalize the raw activations within each subject (subtract mean and divide by standard deviation) across all image responses - max and avg conditions, both natural and synthetic sources and all three target regions. The updated Figure 2 is shown below.

We also added Supplementary Figure 5 to the Supplementary Information, which shows the raw individual fMRI response magnitudes, colored by subject.

Finally, we have added the complete activation by target region plots in the Supplementary Information, attached below as Supplementary Figure 6-8:

Reviewer Comment

The claim that higher-level regions show greater specificity of the optimized stimuli requires both statistical evidence (i.e., testing a region x personalization interaction, not merely comparing region-related p-values) and a clear link to the existing literature on this topic.

Response

We have further analyzed this observation by fitting an LME model to quantify the relationship between the optimized synthetic images' ability for personalization and the hierarchical level of the face perception regions. Specifically, we took the t-statistic between the responses of "Self" and "Group"/"Other" as the dependent variable, the level of the regions (OFA=1, FFA1=2, FFA2=3, mTLfaces=4, aTLfaces=5) and the contrast type ("Self" vs "Group", "Self" vs "Other") as fixed effects and subjects as random effect. We find that there is a positive (although not significant) coefficient for the region level ($\beta = 0.216$, $p = 0.098$), which suggests there is a weak trend toward higher-level regions showing greater specificity of the personalized synthetic images. We also provided some references to the existing work in the Discussion as shown below:

Figure 2: Brain activations in response to the "Max" condition images are significantly higher than the activations in response to the "Avg" condition images for their targeted region (and for almost all regions in the same response category), for both natural ("Nat") and synthetic ("Syn") image sets. Violinplots show the distribution of the normalized fMRI activity in response to the different image conditions (z-scored over all image responses) in **a** face regions, **b** body regions and **c** word regions. Different subjects are represented by different color points. The following group comparisons were performed via fitting linear mixed effects (LME) models: "GroupMaxNat" vs "GroupAvgNat", "GroupMaxSyn" vs "GroupAvgSyn", "GroupMaxSyn" vs "GroupMaxNat" and "GroupAvgSyn" vs "GroupAvgNat". Significant differences based on permutation testing (FDR corrected $p < 0.05$) are marked with a starred horizontal line.

Supplementary Figure 5: Brain activations in response to the "Max" condition images are significantly higher than the activations in response to the "Avg" condition images for their targeted region (and for almost all regions in the same response category), for both natural ("Nat") and synthetic ("Syn") image sets. Scatterplots show the individual observations of the raw fMRI activity in response to the different image conditions in **a** face regions, **b** body regions and **c** word regions. Different subjects are represented by different color points. The following group comparisons were performed via fitting linear mixed effects (LME) models: "GroupMaxNat" vs "GroupAvgNat", "GroupMaxSyn" vs "GroupAvgSyn", "GroupMaxSyn" vs "GroupMaxNat" and "GroupAvgSyn" vs "GroupAvgNat". Significant differences based on permutation testing (FDR corrected $p < 0.05$) are marked with a starred horizontal line.

Responses of FFA1-targeted images in body and word regions

Supplementary Figure 6: Responses of FFA1-targeted images in **a** body and **b** word regions.

Responses of EBA-targeted images in face and word regions

Supplementary Figure 7: Responses of EBA-targeted images in **a** face and **b** word regions.

Responses of VWFA1-targeted images in body and face regions

Supplementary Figure 8: Responses of VWFA1-targeted images in **a** body and **b** face regions.

We observed increasing "success" of the personalized synthetic images when arranging the results from low to high-order (OFA, FFA1, FFA2, mTLfaces to aTLfaces), though the targeted region is only FFA1. To formally test this observation, we performed a post-hoc LME analysis with the t-statistic between "Self" and "Group"/"Other" as the dependent variable, hierarchical level of region (OFA=1, FFA1=2, FFA2=3, mTLfaces=4, aTLfaces=5) and the contrast type ("Self" vs "Group", "Self" vs "Other") as fixed effects, and subjects as random effect. We found a positive coefficient for region hierarchy level, with p -values trending toward significance ($p \sim 0.10$). As all the regions belong to the same perception group with face as their specialization, it is not surprising that images activate FFA1 can also activate other regions. The stronger personalization effect could perhaps be explained by the fact that there is more homogeneity in brain responses across the population to low-level characteristics of face images, like facial topology (Henriksson et al., 2015), but less homogeneity in brain responses to higher-order characteristics, like facial recognition (Barense et al., 2010; Yang et al., 2016).

Reviewer Comment

As I understand it, the described linear mixed effect model does not consider images as a random factor. In fact, this MLE appears to be a degenerate case equivalent to a paired t-test across subjects, limiting the general applicability of the results.

Response

Because our variable of interest was the overall response to the "max" and "avg" image conditions for each subject, this was a fixed effect. No images are shared across the conditions, each image is unique and it would thus not make sense to include image identity as a random factor in the model.

Reviewer Minor Comment

The fact that AlexNet, trained on ImageNet, was used as the basis for the encoding model should be mentioned at the onset of the results section. The absence of model specifics from the results is curious.

Response

We have added more details about the encoding model at the beginning of the Results section.

We created eight individual-level encoding models for the above three regions using a deepnet feature-weighted receptive field (deepnet-fwRF) architecture (St-Yves and Naselaris, 2018). The deepnet-fwRF model has an ImageNet-pretrained AlexNet (Krizhevsky et al., 2012) to extract features, a Gaussian pooling field to reduce the number of the features and a Ridge regression to map the features to brain regional response, see details in the Methods section.

Reviewer Minor Comment

A proper permutation test p-value cannot be zero (Phipson & Smyth, 2010 doi: 10.2202/1544-6115.1585).

Response

We have changed the text to "approximately 0".

Reviewer Minor Comment

The statement, "human experiments obviously must rely on non-invasive recording techniques," does not take electrocorticography into account.

Response

This is true - thank you. We have changed modified the sentence in question:

Unlike experiments in macaque monkeys where microelectrode arrays can be invasively implanted directly on the brain to record neuronal responses, human experiments mostly rely on non-invasive recording techniques, with the exception of electrocorticography which is only used in small cohorts of neurological patients (Haufe et al., 2018).

Reviewer Minor Comment

The phrase, "Then the number of features in each AlexNet layer were reduced by selecting those that had the highest variance," is ambiguous. Please precisely describe the feature selection procedure and the method for its cross-validation.

Response

We have added more details as below:

The maximum number of feature maps in each AlexNet layer is set to 512. For layers that have more than 512 feature maps, we selected the top 512 maps that had the highest variance across the whole NSD image set. Then feature maps that have the same spatial resolution were concatenated. A Gaussian pooling field was applied to the feature maps to further reduce the number of features before the final ridge regression which mapped the features to brain regions' responses, which is the average of the voxel-wise activation maps over that region (a scalar). The hyperparameters, namely the center and radius of the Gaussian pooling field, and the regularization parameter of the Ridge regression, were determined by choosing the combination that gave the best performance on a held-out validation set of 3000 image-response pairs using grid search.

References

- Barense, M. D., Henson, R. N., Lee, A. C., and Graham, K. S. (2010). Medial temporal lobe activity during complex discrimination of faces, objects, and scenes: Effects of viewpoint. *Hippocampus*, 20(3):389–401.
- Gu, Z., Jamison, K. W., Khosla, M., Allen, E. J., Wu, Y., Naselaris, T., Kay, K., Sabuncu, M. R., and Kuceyeski, A. (2022). Neurogen: activation optimized image synthesis for discovery neuroscience. *NeuroImage*, 247:118812.
- Haufe, S., DeGuzman, P., Henin, S., Arcaro, M., Honey, C. J., Hasson, U., and Parra, L. C. (2018). Elucidating relations between fmri, ecog, and eeg through a common natural stimulus. *NeuroImage*, 179:79–91.
- Henriksson, L., Mur, M., and Kriegeskorte, N. (2015). Faciotopy—a face-feature map with face-like topology in the human occipital face area. *Cortex*, 72:156–167.
- Krizhevsky, A., Sutskever, I., and Hinton, G. E. (2012). Imagenet classification with deep convolutional neural networks. *Advances in neural information processing systems*, 25:1097–1105.
- St-Yves, G. and Naselaris, T. (2018). The feature-weighted receptive field: an interpretable encoding model for complex feature spaces. *NeuroImage*, 180:188–202.
- Yang, H., Susilo, T., and Duchaine, B. (2016). The anterior temporal face area contains invariant representations of face identity that can persist despite the loss of right ffa and ofa. *Cerebral Cortex*, 26(3):1096–1107.

Reviewers' comments:

Reviewer #1 (Remarks to the Author):

I'd like to thank the authors for their thorough revisions.

From a scientific results standpoint, I think the paper can be accepted as-is. That being said, I would suggest that the authors continue to make edits for ease of reading/understanding. While I think the paper is substantially more readable/understandable than the previous version, I still found aspects of the Results section challenging to get through.

I think the section "Observed and targeted brain activation patterns are well aligned" is a good example. It is a pretty long paragraph, with many different results listed throughout the paragraph, without clear motivation of analyses at the beginning, or synthesis of findings at the end. I know there are many different styles of scientific writing, but for easiest readability I would personally recommend, throughout the results section, to 1) split up long paragraphs so that paragraphs relate to a single topic/scientific question; and 2) have motivation at the beginning of the paragraph and/or a synthesis at the end of the paragraph. Ultimately, the writing style is up to you (I don't think reviewers should accept/reject based on that), but I wanted to share my two cents about how I think the paper could be improved to better get your scientific message across to readers.

A few additional very minor comments:

-In Fig. 1, many brain region acronyms are listed in the caption that aren't in the figure. Most of these acronyms would be helpful instead in the caption of Fig. 2

-When you introduce the LME on page 2, I think it would be helpful to readers if you mention what is the fixed-effect and random effect, as you do on page 6.

-In the second paragraph of results, you say "For natural image selection, the candidate natural images set were the 9,000 \times 8 = 72,000 images shown to any one of the eight NSD subjects that did not belong to the shared 1000 image set." You did not previously mention what the "shared 1000 image set" is, so this is confusing.

Reviewer #2 (Remarks to the Author):

I am satisfied with the revised manuscript. Thanks to the authors for their efforts.

Reviewer #3 (Remarks to the Author):

I appreciate the authors' significant efforts in addressing the comments from both myself and other reviewers. I am particularly grateful for the sharing of code and stimuli. I am looking forward to seeing the manuscript in its published form. Below, I list remaining points for further improving it before it is published.

Data Sharing

One major issue remains regarding data sharing. The manuscript states, "The NeuroGen Dataset will be made available upon reasonable request." This stance does not align with best practices in open science. I recommend that the authors provide a link to the figshare page where they plan to share the fMRI results or, alternatively, to openneuro. Although the IRB might have reservations about

sharing complete MRI data, sharing region-level response summaries (shaped as a tensor with dimensions of ROI X IMAGE X SUBJECT X HEMISPHERE X REPEATS) should not pose privacy risks. I noticed some activation `.npy` files in the stimulus set on figshare, but these do not include the complete activation tensor as described above. Including both standardized and raw BOLD activity measures would be beneficial.

Additionally, I recommend that the fitted encoding models be shared to facilitate the reproduction of the stimulus synthesis process by other researchers.

Stimuli Files

Some folders in the stimuli files on figshare are titled "randnat" and "randsyn." Do these relate to the AVG conditions? A clear readme file to explain the dataset would be valuable.

Code Repository

- The code seems to import a module not included in the GitHub repository: `from torchmodel.models.alexnet import Alexnet_fmmaps``.
- Is there code available for stimulus selection? I only saw code related to synthesis.

Overall, a functional and complete repository would substantially enhance the impact of this work.

Random Effects

Regarding the treatment of stimuli as a random factor, it is not necessary for the stimuli to be shared across conditions to qualify as a random factor. This is an example of a *nested random factor*. To illustrate, in school-based experiments, pupils and classes serve as analogs for images and conditions, respectively.

In an experiment with a single subject and multiple images across two experimental conditions, there would be one fixed factor (*condition*) and one random factor (*image*). In another scenario, where each participant is tested on one image per condition, the random factor would be the *subject* and the fixed factor would be the *condition*. Given that your experiment involves multiple participants and images, the most comprehensive model would include a fixed factor for condition and random factors for both images and subjects. I recommend that the authors either revise their analysis to include images as a random factor or explicitly state the limitation of the current model: its inferences generalize beyond the sample of subjects but not beyond the sample of images.

Correlation Across ROIs

For evaluating correlation across ROIs, using Cohen's D may be more conceptually appropriate than a T-score, although the numerical result should remain the same. This would remove the influence of sample size from the denominator.

Deepnet Feature Weighted Receptive Field Encoding Model

- The statement "we selected the top 512 maps that had the highest variance across the whole NSD image set" is unclear. Did you first average each map across space, reducing it to a scalar per image, before calculating the variance across images? Please clarify.
- The sentence "feature maps that have the same spatial resolution were concatenated" implies that

you might have flattened the 2D maps into 1D vectors, which is inconsistent with the subsequent application of FWRP. Could you specify the hyper-parameters for the FWRP grid search?

Units

The newly standardized units are an improvement but still pose some interpretative challenges. Consider adopting a percent change relative to the BOLD baseline level or the AVG condition for better clarity. If you opt to retain the current units, at least include the raw BOLD measurements in the shared data.

Repetitions

Does the manuscript explicitly describe the number of repetitions (i.e., presentations) for each image? I might have missed this detail.

Response to Reviewers

**Title: Modulating human brain responses via optimal natural image selection
and synthetic image generation**

**Manuscript Reference Number:
COMMSBIO-23-1582A**

Authors:

Zijin Gu

Keith Jamison

Mert Sabuncu

Amy Kuceyeski

Date: September 16, 2023

Message from the Authors

Dear Reviewers,

We thank the reviewers for their constructive comments, which have allowed us to improve the quality of the manuscript. We have addressed the comments and incorporated these valuable suggestions in the current manuscript which is a substantial revision; we believe that the result is a strong manuscript that will be of great interest to the neuroscientific community, vision researchers in particular. The updated contents are colored in blue in the revised manuscript to indicate our changes.

We have made many changes, including addressing all statistical concerns and discussing any remaining limitations, made publicly available the processed regional BOLD responses for both experiments and all subjects, adding discussion related to the underlying models used in the study, and reformatting the text to increase the clarity of the presentation. With regards to the editorial comment regarding further elaboration on the novelty of this study over our previous work (Reference 24), we have added text to both the Introduction and Discussion (Intro: "In Gu et al. (2022), we validated our approach for building personalized encoding models using small amounts of data, but did not test our framework for targeted activation of given brain regions in specific individuals, which is what we present here.", Discussion: "In addition, inter-individual variability of responses in face regions was considered when creating/selecting the optimal images using an encoding model approach we developed and validated previously Gu et al. (2022). Here, we showed that personalization did indeed drive responses for specific individuals above and beyond the responses to images designed using a group-level encoding model or other individuals' encoding models, but only for synthetic images and only in face regions that were higher in the processing hierarchy."). All page and figure numbers in our response are based on the revised manuscript, unless otherwise stated. The page and reference numbers mentioned in the reviewers' comments are kept intact and are based on the original manuscript. The references contained in this reviewer response document are in author-year format for ease of reading, and are listed at the end of this document. Thank you and we look forward to hearing the journal's decision.

Sincerely,

Zijin Gu, Keith Jamison, Mert Sabuncu, Amy Kuceyeski

Response To Reviewer #1

Overall Comments

I'd like to thank the authors for their thorough revisions. From a scientific results standpoint, I think the paper can be accepted as-is. That being said, I would suggest that the authors continue to make edits for ease of reading/understanding. While I think the paper is substantially more readable/understandable than the previous version, I still found aspects of the Results section challenging to get through.

I think the section "Observed and targeted brain activation patterns are well aligned" is a good example. It is a pretty long paragraph, with many different results listed throughout the paragraph, without clear motivation of analyses at the beginning, or synthesis of findings at the end. I know there are many different styles of scientific writing, but for easiest readability I would personally recommend, throughout the results section, to 1) split up long paragraphs so that paragraphs relate to a single topic/scientific question; and 2) have motivation at the beginning of the paragraph and/or a synthesis at the end of the paragraph. Ultimately, the writing style is up to you (I don't think reviewers should accept/reject based on that), but I wanted to share my two cents about how I think the paper could be improved to better get your scientific message across to readers.

Response

Thank you for your recognition of our revision. We appreciate your valuable suggestions on the styles of scientific writing, and we made some more modifications accordingly. We hope you find the revised manuscript easier to read and follow. Please also see our point-by-point responses below.

Reviewer Comment

In Fig. 1, many brain region acronyms are listed in the caption that aren't in the figure. Most of these acronyms would be helpful instead in the caption of Fig. 2

Response

We moved the acronyms that are not used in Fig 1 to Fig 2 caption.

Reviewer Comment

When you introduce the LME on page 2, I think it would be helpful to readers if you mention what is the fixed-effect and random effect, as you do on page 6.

Response

We added the fixed effect and random effect to the results on page 2 as follows:

We fit a linear mixed effects (LME) model with different image conditions as fixed effect and subjects as random effect, and compared the brain activations between image conditions ("GroupMaxNat", "GroupAvgNat", "GroupMaxSyn", "GroupAvgSyn") for all six subjects in the three primary target regions, see Figure 2a.

Reviewer Comment

In the second paragraph of results, you say "For natural image selection, the candidate natural images set were the $9,000 \times 8 = 72,000$ images shown to any one of the eight NSD subjects that did not belong to the shared 1000 image set." You did not previously mention what the "shared 1000 image set" is, so this is confusing.

Response

We agree "shared 1000 image set" is not clear, so we change it to

For natural image selection, the candidate natural images set were the $9,000 \times 8 = 72,000$ images shown to any one of the eight NSD subjects while the 1000 images that were shared across subjects were not included.

Response To Reviewer #2

Overall Comments

I am satisfied with the revised manuscript. Thanks to the authors for their efforts.

Response

Thank you for your recognition of our revised manuscript.

Response To Reviewer #3

Overall Comments

I appreciate the authors' significant efforts in addressing the comments from both myself and other reviewers. I am particularly grateful for the sharing of code and stimuli. I am looking forward to seeing the manuscript in its published form. Below, I list remaining points for further improving it before it is published.

Response

Thank you for your recognition of our revision. Your suggestions have greatly helped improve our work and please find the point-to-point responses below.

Reviewer Comment

Data Sharing

One major issue remains regarding data sharing. The manuscript states, "The NeuroGen Dataset will be made available upon reasonable request." This stance does not align with best practices in open science. I recommend that the authors provide a link to the figshare page where they plan to share the fMRI results or, alternatively, to openneuro. Although the IRB might have reservations about sharing complete MRI data, sharing region-level response summaries (shaped as a tensor with dimensions of ROI X IMAGE X SUBJECT X HEMISPHERE X REPEATS) should not pose privacy risks. I noticed some activation '.npy' files in the stimulus set on figshare, but these do not include the complete activation tensor as described above. Including both standardized and raw BOLD activity measures would be beneficial.

Additionally, I recommend that the fitted encoding models be shared to facilitate the reproduction of the stimulus synthesis process by other researchers.

Response

Our IRB finally approved to sharing of processed regional level brain responses for two sessions, and we have updated the figshare repository to contain this data. However, we are still unable to publish the raw fMRI data due to the restrictions from IRB. We also uploaded the fitted encoding models for completeness. Here is our updated data repository https://figshare.com/articles/dataset/NeuroGen_Dataset/23582403 and code repository <https://github.com/zijin-gu/neural-modulation>.

Reviewer Comment

Stimuli Files

Some folders in the stimuli files on figshare are titled "randnat" and "randsyn."
Do these relate to the AVG conditions? A clear readme file to explain the dataset would be valuable.

Response

We added a readme file to the figshare dataset. To answer your query, the "randnat" is the AVG natural condition and "randsyn" is the AVG synthetic condition. We think this is made clear in the readme.

Reviewer Comment

Code Repository

- The code seems to import a module not included in the GitHub repository: "from torchmodel.models.alexnet import Alexnet_fmmaps".
- Is there code available for stimulus selection? I only saw code related to synthesis.

Overall, a functional and complete repository would substantially enhance the impact of this work.

Response

- Thank you for the reminder. We added the required code to the github repository.
- We selected stimuli based on the encoding model predictions. Once the data and the model are ready, selection can be done by feeding the images to the models and collecting the predictions. And we uploaded a example script for selecting individual images here https://github.com/zijin-gu/neural-modulation/blob/main/select_individual_images.py.

We have a completed pipeline of NeuroGen which can be found here <https://github.com/zijin-gu/NeuroGen>, which contains neural encoding model training and synthetic image generation.

Reviewer Comment

Random Effects

Regarding the treatment of stimuli as a random factor, it is not necessary for the stimuli to be shared across conditions to qualify as a random factor. This is an example of a nested random factor. To illustrate, in school-based experiments, pupils and classes serve as analogs for images and conditions, respectively.

In an experiment with a single subject and multiple images across two experimental conditions, there would be one fixed factor (condition) and one random factor

(image). In another scenario, where each participant is tested on one image per condition, the random factor would be the subject and the fixed factor would be the condition. Given that your experiment involves multiple participants and images, the most comprehensive model would include a fixed factor for condition and random factors for both images and subjects. I recommend that the authors either revise their analysis to include images as a random factor or explicitly state the limitation of the current model: its inferences generalize beyond the sample of subjects but not beyond the sample of images.

Response

In the LME model, we want to ask whether there is significant difference between two image conditions, where multiple images are drawn from some set that all satisfy the condition in question. We are not interested in differences that exist in responses to a particular image; rather the differences that exist across conditions. In your analogy of classes/pupils, we are interested in the differences across classes, not the pupils in them (which are not shared across classes). We agree that one limitation of the current model is that it may be difficult to generalize this effect for different images, and we have added the limitation to the discussion

"Finally, since we were interested in the effect of the image condition (and not the individual images themselves), we did not include image as a random effect; this could mean that the results may not generalize to out-of-sample images."

Reviewer Comment

Correlation Across ROIs

For evaluating correlation across ROIs, using Cohen's D may be more conceptually appropriate than a T-score, although the numerical result should remain the same. This would remove the influence of sample size from the denominator.

Response

We have updated the t-statistics in Figure 3 to rather include Cohen's D; the results are unchanged.

Reviewer Comment

Deepnet Feature Weighted Receptive Field Encoding Model

- The statement "we selected the top 512 maps that had the highest variance across the whole NSD image set" is unclear. Did you first average each map across space, reducing it to a scalar per image, before calculating the variance across images? Please clarify.
- The sentence "feature maps that have the same spatial resolution were concatenated" implies that you might have flattened the 2D maps into 1D vectors, which is inconsistent with the subsequent application of FWRP. Could you

specify the hyper-parameters for the FWRP grid search?

Response

We modified our descriptions of the encoding model construction based on your comments, and add additional explanations below.

- We first calculate the variance of each of the feature maps across all images in NSD, and then take the average variance over the images to obtain a scalar variance score for each feature map. We then sort the feature maps based on the variance score and take the top 512 ones. We have modified the statement as pasted below.

For layers that have more than 512 feature maps, we calculated the variance of each of the feature maps for all images in NSD and then selected the top 512 maps that had the highest average variance across the images.

- We didn't flatten 2D feature maps. As we wrote in the paper, we only concatenate feature maps of the same spatial resolution. To be clear, we added more details of the feature maps as below.

Then feature maps that have the same spatial resolution were concatenated, which resulted in three concatenated feature maps with size (256, 27, 27), (896, 13, 13) and (1536, 1, 1).

And we added more details to how the grid search were conducted as below.

Specifically, the candidate feature pooling field centers were spaced 1.4 degrees apart, the candidate radius included 8 log-spaced receptive field sizes between 0.04 and 0.4, and the candidate regularization parameters were 9 log-spaced values between $10^3 \sim 10^7$.

Reviewer Comment

Units

The newly standardized units are an improvement but still pose some interpretative challenges. Consider adopting a percent change relative to the BOLD baseline level or the AVG condition for better clarity. If you opt to retain the current units, at least include the raw BOLD measurements in the shared data.

Response

The responses were normalized in a way that provides a clear visualization of the violin plots; it is difficult to interpret the results when grouping subjects together that all have slightly different response magnitudes. When doing the comparisons, we used the un-normalized raw BOLD response data (which we now share freely with the code that accompanies this publication).

In our previous version of the paper, we normalized with respect to the AVG condition. However, that reduces the meaning in the AVG responses. It was flagged previously by a reviewer and we agreed - thus we have the current normalization scheme.

Reviewer Comment

Repetitions

Does the manuscript explicitly describe the number of repetitions (i.e., presentations) for each image? I might have missed this detail.

Response

Yes we wrote it in the Method section and also pasted below for your reference

8 unique stimuli were presented per block, with one image repeated in each block for use as a one-back behavioral task.

References

Gu, Z., Jamison, K., Sabuncu, M., and Kuceyeski, A. (2022). Personalized visual encoding model construction with small data. *Communications Biology*, 5(1):1382.